# Intercavity polariton slows down dynamics in strongly coupled cavities

Yesenia A. García Jomaso[1], Brenda Vargas[1], David Ley Domínguez [1], Román J. Armenta-Rico [1], Huziel E. Sauceda [1], César L. Ordoñez-Romero [1], Hugo A. Lara-García [1], Arturo Camacho-Guardian [1] ✉ & Giuseppe Pirruccio [1] ✉

Band engineering stands as an efficient route to induce strongly correlated quantum many-body phenomena. Besides inspiring analogies among diverse physical fields, tuning on demand the group velocity is highly attractive in photonics because it allows unconventional flows of light. Λ-schemes offer a route to control the propagation of light in a lattice-free configurations, enabling exotic phases such as slow-light and allowing for highly optical non-linear systems. Here, we realize room-temperature intercavity Frenkel polaritons excited across two strongly coupled cavities. We demonstrate the formation of a tuneable heavy-polariton, akin to slow light, appearing in the absence of a periodic in-plane potential. Our photonic architecture based on a simple three-level scheme enables the unique spatial segregation of photons and excitons in different cavities and maintains a balanced degree of mixing between them. This unveils a dynamical competition between many-body scattering processes and the underlying polariton nature which leads to an increased fluorescence lifetime. The intercavity polariton features are further revealed under appropriate resonant pumping, where we observe suppression of the polariton fluorescence intensity.

The ability to tune band dispersion is a powerful means for controlling the competition between kinetic energy and interactions, crucial to inducing strongly correlated phases. In condensed matter, the advent of new quantum materials enriched the platform for band engineering[1], permitting the realization of intriguing phases such as unconventional superconductivity[2–4]. In optics, manipulating band structures unfolded new opportunities to design light-matter interactions and nonlinear optical devices[5,6]. Here, a natural approach involves the design of ordered structures such as photonic crystals[7–11], metamaterials[12,13], and exciton-polariton lattices[14–17].

Another powerful way to realize intriguing forms of light are three-level energy schemes, schematically depicted in Fig. 1a. Despite their apparent simplicity, they are at the core of phenomena such as slow-light in atomic gases[18–20], dipolaritons[21–23], trion-polaritons[24,25],

and stands as a promising avenue for the design and control of novel states of light and matter such as quasiparticles with negative mass[26–29]. Slow-light is achieved through the dramatic suppression of light dispersion[18] and has garnered considerable interest in the fields of quantum optics[30], atomic physics[31,32] and condensed matter[33,34]. In the quantum domain, slow-light is typically understood in terms of polaritons[35], hybrid light-matter quasiparticles that arise from mixing a photon with an elementary matter excitation. Therefore, three-level schemes provide a powerful alternative route for controlling light dispersion with lattice-free setups.

Inspired by the control of light states provided by the Λ-scheme, in this article, we successfully implement intercavity polaritons at room temperature through the strong coupling of a photonic cavity with a polaritonic cavity. The dispersion of the so-formed intercavity

[1]Instituto de Física, Universidad Nacional Autónoma de México, Apartado Postal 20-364, Ciudad de México, C.P. 01000, Mexico.
✉e-mail: acamacho@fisica.unam.mx; pirruccio@fisica.unam.mx

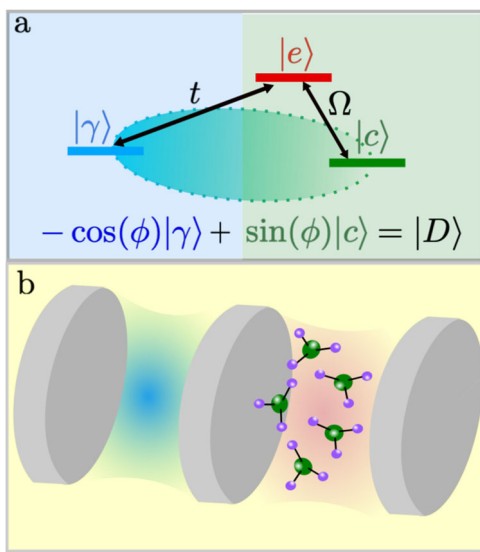

**Fig. 1 | Three-level scheme. a** Generic three-level scheme (in a Λ configuration): A $|\gamma\rangle$ state couples directly to an intermediate state $|e\rangle$. On resonance, the coupling between $|e\rangle$ and a third state $|c\rangle$ gives rise to a hybrid state $|D\rangle$ with vanishing contribution from the intermediate state $|e\rangle$. Here, $\phi$ represents the mixing angle between $|\gamma\rangle$ and $|c\rangle$. **b** Representation of the hybrid photonic-polaritonic coupled system. Here, $|\gamma\rangle = |\omega_c^L\rangle$, $|e\rangle = |\omega_c^R\rangle$ and $|c\rangle = |\omega_c^X\rangle$ represent the left photon, right photon and exciton state, respectively. Photons tunnel between the left and right cavity with a hopping amplitude $t$, while $\Omega$ is the light-matter coupling. In this case, the emergent $|D\rangle$ state corresponds to a pure intercavity polariton.

polaritons is tailored by tweaking the associated three-energy-level diagram. We demonstrate that the middle polariton transits from a hybrid intra- and intercavity polariton to a pure intercavity polariton, a mixture characterized by a stretched polariton lifetime, with a photon and exciton component that are spatially segregated. Under careful resonant pumping a suppressed light emission signals the emergence of a pure intercavity polariton. The indirect nature of intercavity polaritons motivates proposals for quantum tomography protocols. These protocols involve probing the photonic and molecular degrees of freedom through local and independent measurements, such as optical and transport measurements, respectively.

## Results
### Intercavity polaritons
Up until now, the exploration of strongly coupled photonic cavities has primarily centered around photonic crystal cavities[36], micro-rings[37], and whispering-gallery modes[38]. In the context of semiconductor microcavities, double-well potentials have led to the realization of phenomena like Josephson oscillations and self-trapping with intercavity polaritons,[39,40] and have been recently proposed for quantum chemistry control[41].

To realize slow-light and intercavity polaritons we designed a three-level energy scheme representing two strongly coupled cavities. This set-up reminds of the Λ-scheme commonly employed in quantum control experiments. Our system, sketched in Fig. 1b, is composed by two cavities, labeled (Left) and (Right), coupled *via* a thin mirror. The left cavity, represented by $|\omega_c^L\rangle$, is purely photonic, whereas the right cavity hosts both photons and excitons, identified by $|\omega_c^R\rangle$ and $|\omega_c^X\rangle$, respectively. The Hamiltonian of the system is given by

$$\hat{H} = \begin{bmatrix} \omega_c^L(\theta) & -t & 0 \\ -t & \omega_c^R(\theta) & \Omega \\ 0 & \Omega & \omega_X \end{bmatrix}, \qquad (1)$$

where $\omega_c^{L/R}(\theta)$ indicates the energy of the left/right cavity photons, and $\omega_X$ the energy of the excitons. Here, the angle $\theta$ corresponds to the incident angle of light injected into the left cavity. In Fig. 1b, the photon hopping is characterized by a tunneling amplitude $t$, while the light-matter coupling is given by the Rabi frequency, $\Omega$.

The sample consists of two vertically stacked nanocavities fabricated on a glass substrate by a sequence of multiple sputtering and spin-coating steps. The front, middle, and back mirrors are made by Ag and their thickness equal to 20 nm, 20 nm and 300 nm, respectively. The middle mirror width determines the tunneling amplitude $t$. The Left nanocavity is filled with polymethyl methacrylate (PMMA), whereas the Right one embeds a dye-doped polyvinyl alcohol (PVA) layer. The excitonic content is provided by a high concentration of homogeneously dispersed Erythrosin B (ErB) molecules[42]. The absorption spectrum of the active medium exhibits a main peak around $\omega_X \approx 2.24$ eV associated to a principal exciton resonance and a second peak at $\omega_\nu \approx 2.4$ eV related to the first vibron mode. The polymer thickness of the Left cavity features a slow wedge which provides us the possibility of fine tuning $|\omega_c^L\rangle$ in a wide photon energy range.

On resonance, when the frequency of the left cavity at normal incidence matches the exciton one, the middle polariton becomes

$$|D(\theta=0)\rangle = \frac{\Omega}{\sqrt{\Omega^2+t^2}}|\omega_c^L(\theta=0)\rangle + \frac{t}{\sqrt{\Omega^2+t^2}}|\omega_X\rangle, \qquad (2)$$

that is, a state solely formed by the superposition of the left cavity photon and the exciton. We refer to this state as a pure intercavity polariton. Interestingly, the exciton and photon forming the polariton state are spatially separated and the right cavity photon does not participate in the formation of this polariton branch. For intercavity polaritons, the mixing angle $\phi$ (Fig. 1a) is given by $\sin(\phi) = t/\sqrt{\Omega^2+t^2}$ which depends solely on the Rabi coupling $\Omega$ and the tunneling coefficient $t$.

The character of the pure intercavity polariton, obtained from the diagonalization of Eq. (1), reminds of the dark-state polariton observed in slow-light experiments performed with atomic gases[20]. In atomic systems such light-matter mixture can be achieved by employing atoms with internal energy levels resembling the Λ scheme shown in Fig. 1a. In that case, a photon, $|\gamma\rangle$, directly couples to an atomic excited short-lived state, $|e\rangle$, while a classical field couples the atomic short-lived state to a different, metastable atomic state, $|c\rangle$. Subsequently, a polariton, $|D\rangle$, forms through the hybridization of the photon and the metastable state. This phenomenon can be understood as an interference effect that allows light to propagate undamped in an otherwise opaque medium. As transparency is induced by the classical light field, $\Omega$, this phenomenon is coined electromagnetically induced transparency (EIT). $|D\rangle$ is commonly referred to as a dark-state polariton. Our intercavity polariton in coupled photonic-polaritonic cavities is the analog of a dark-state polariton in atomic gases. we stress that it is essential to distinguish the concept of a dark-state polariton in atomic gases - a form of light that propagates undamped - from the dark-state terminology employed in condensed matter, where, by definition, a dark state is decoupled from light.

In the following, we experimentally unveil the character of the polariton states. Our reflectance and photoluminescence measurements will offer insights into the spectral composition of the polaritons, including energy bands, bandwidths, and Hopfield coefficients, which will allow us to demonstrate the formation of pure intercavity polaritons. On the other hand, lifetime measurements give us comprehension of the polariton dynamics. These complementary signatures provide a deep understanding of our results.

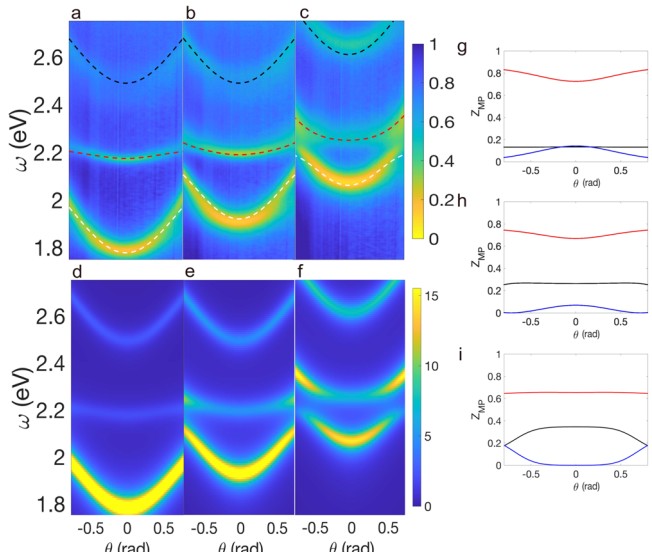

**Fig. 2 | Intercavity polaritons.** s-polarized reflectance as a function of the angle of incidence and photon energy for **a** $\delta/eV = -0.37$, **b** $\delta/eV = -0.18$ and **c** $\delta/eV = 0$. The white, red, and black dashed curves correspond to the theoretical fitting of the energies of the lower, middle, and upper polariton respectively. **d**–**f** Spectral function, $A^L(\theta, \omega)$, calculated for the same detunings as in **a**–**c**. Middle polariton Hopfield coefficients for **g** $\delta/eV = -0.37$, **h** $\delta/eV = -0.18$ and **i** $\delta/eV = 0$. The black, blue, and red curves correspond to the left cavity photon, right cavity photon, and exciton component, respectively.

## Polariton spectrum

In Fig. 2a–c, we show the local band structure of the coupled cavity system measured via Fourier microscopy, which demonstrates that the Λ-scheme yields a middle polariton (MP) state with energies close to the bare exciton one. Fine-tuning of the MP energy is accomplished by leveraging the wedged PMMA thickness. The MP state shows a reduced dispersive character compared to the upper (UP) and lower polariton (LP) and flattens for a specific value of the thickness of the photonic cavity. From Eq. (2) we expect the pure intercavity polariton to be visible when light is injected from the left cavity, whereas reflectance measurements should not reveal this state when photons are pumped from the right one. For more details, please refer to the Supplementary Information.

To theoretically understand our results from the polariton perspective, the simple picture of three quantum coupled oscillators in Eq. (1) suffices. Here, the energies and Hopfield coefficients are given by the matrix eigenvalues and eigenvectors, respectively. We introduce the spectral function

$$A^L(\theta, \omega) = -2\text{Im} \sum_\alpha \frac{|\langle \alpha | \omega_c^L(\theta) \rangle|^2}{\omega - \omega_\alpha(\theta) + i\gamma_\alpha}, \tag{3}$$

where the states $|\alpha\rangle$ correspond to the three polariton branches (LP, MP, UP) and $\gamma_\alpha$ denote the linewidths of the polariton branches[43]. Here, $A^L(\theta, \omega)$ provides the spectral composition of the left cavity photons in terms of the three polariton branches, allowing us to directly compare with our experiment. The spectral function is shown in Fig. 2d–f for three values of the detuning $\delta = \omega_c^L - \omega_X$ and allows us to compare the energies, the linewidths, as well as the spectral composition of polariton states. We obtain a very good agreement with the experimentally observed reflectance. By continuously varying the left cavity thickness, its photon energy is driven in resonance with the exciton energy. Furthermore, the energy of the polariton states obtained from our theory and plotted with dashed curves on the reflectance maps, Fig. 2a–c, provide an excellent quantitative understanding of the system.

We obtain a reduction of the polariton dispersion as

$$\omega_{\text{MP}}(\theta) \approx \omega_X + \left[ \omega_c^L(\theta) - \omega_X \right] \frac{1}{1 + \left(\frac{t}{\Omega}\right)^2}. \tag{4}$$

This means that the dispersion of the MP is controlled by the tunneling ratio, $t$, which can be tailored by means of the middle mirror thickness. On resonance, the MP becomes a pure intercavity polariton with an energy matching the bare exciton one, its dispersion is reduced by a factor 4–8 compared to the upper and lower polaritons.

The character of the MP is clearly unveiled once it is written in terms of the bare photon and exciton states, i.e., $|\text{MP}\rangle = \sum_i \mathcal{C}_{\text{MP}}^i |i\rangle$. The amplitude of its Hopfield coefficients $Z_{\text{MP}}^i = |\mathcal{C}_{\text{MP}}^i|^2$ in Fig. 2g–i demonstrates that the MP decouples from the right cavity photons for $\delta = 0$. Furthermore, Fig. 2i shows that the Hopfield coefficients remain invariant along the angular region where the MP is a heavy pure intercavity polariton. There, the MP has only non-vanishing Hopfield coefficients of the left cavity photon and the exciton.

This suppressed curvature is retained over a wide range of incident angles. However, for $\delta/eV = 0$ and large angles, it becomes considerably larger than for $\delta/eV = -0.37$. To understand this, it is important to stress that the dispersion of the MP band can be arbitrarily reduced and strongly suppressed with large detunings. However, this completely breaks the pure intercavity nature of the polariton and places the MP energy far away from the bare exciton energy. In our measurements, this effect can be seen for $\delta/eV = -0.37$, where the MP dispersion around normal incidence is small and its Hopfield coefficients confirm that the curvature reduction is achieved at the cost of ceding its left photonic component and becoming a mixed inter-intra cavity polariton. See Supplementary Information S4. Emergence of heavy-mass intercavity polaritons, for additional reflectance and PL measurements at different cavity detunings.

For conventional polaritons arising in two-level schemes, the dispersion of the polariton bands depends on the cavity detuning from the excitons and cannot be further tuned. Suppression of this dispersion can only be achieved by compromising the degree of mixing between photons and excitons. In contrast, the versatility of our three-level scheme provides a mechanism to tune the polariton dispersion while retaining a significant photonic component (See Supplementary Information S2. Two-level vs Three-level polaritons for a detailed comparison).

The intercavity polariton is robust against the detuning of the right cavity from the exciton, denoted as $\delta_{RX} = \omega_c^R - \omega_X$. In fact, its emergence and all of its quasiparticle properties are completely independent of $\delta_{RX}$ and only depend on $t$ and $\Omega$. Away from this condition, the pure intercavity nature of the MP breaks down, and the quasiparticle features start depending on $\delta_{RX}$ (See Supplementary Information S3. The role of the right cavity detuning).

As discussed previously, the formation of the pure intercavity polariton can be understood in terms of the interference between different excitation pathways within the Λ-scheme, as shown in Fig. 3a. Its phenomenology has been extensively studied in the context of atomic gases[19,20], for the realization of dipolar interaction with microcavity polaritons[21–23,44], and for the study of trion-polaritons[24,25].

## Short-time dynamics

Now we focus on the effect of the pure intercavity nature of the MP, hinting at a mechanism analogous to the coherent population trapping observed in atomic electromagnetically induced transparency. For this, we measure the prompt fluorescence decay employing the time-correlated single-photon counting technique. We explore the dynamics of the LP and MP and relate them with the corresponding photon component. The coupled cavities are pumped off-resonantly with laser pulses centered around 2.42 eV. Figure 3b displays decays for two detunings used in Fig. 2, i.e., $\delta/eV = 0$ and $\delta/eV = -0.37$. In the

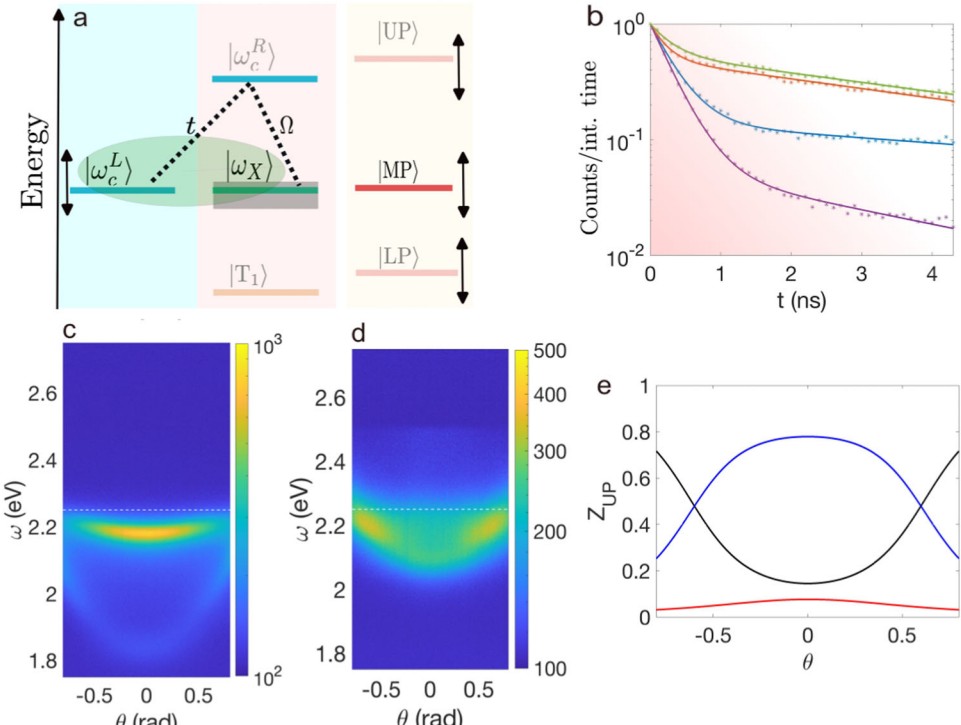

**Fig. 3 | Short-time dynamics and intercavity polaritons. a** Sketch of the relevant energy levels. Fine-tuning the left cavity photon energy shifts the energy of the three polariton states and leads to a pure intercavity polariton, $|MP\rangle$, formed by the admixture of $|\omega_c^L\rangle$ and $|\omega_X\rangle$ (green shaded area). The gray area symbolizes the presence of the dark exciton reservoir. **b** Normalized fluorescence lifetime showing the short-time dynamics of the middle and lower polaritons. The dynamics of the MP is displayed with orange and green asterisks for $\delta/eV = -0.37$ and $\delta/eV = 0.0$, respectively. The LP decay is illustrated with blue and purple asterisks for $\delta/eV = -0.37$ and

$\delta/eV = 0$, respectively. Shaded pink depicts the region where the fastest decay component dominates the early polariton dynamics. **c, d** s-polarized fluorescence, expressed in counts per integration time, for $\delta/eV = -0.37$ and $\delta/eV = 0$, respectively. The energy of the bare exciton is represented by the dashed white line. Photons are injected at $\omega_p/eV = 2.6$ which lies at the energy of the upper polariton. **e** Hopfield coefficients of the UP for $\delta/eV = 0$ showing that it is predominantly formed by right cavity photons. The black, blue and red curves correspond to the left cavity photon, right cavity photon and exciton component, respectively.

considered time interval, all decays are well described by two time constants. For our analysis, we focus initial dynamics illustrated in the pink shaded region. We model the evolution assuming a dynamical equation of the form $C(t) = A \exp(-\Gamma t) + (1 - A) \exp(-\eta t)$, normalized to $C(t = 0) = 1$. Figure 3a shows that this model (solid curves) provides a very good fit to the experimental data (points). We obtain that the LP dynamics is dominated by a single exponential with $\Gamma_{LP}(\delta = 0) = 3.25\,\text{ns}^{-1}$ and $A_{LP} = 0.94$, while $\Gamma_{LP}(\delta = -0.37) = 3.0\,\text{ns}^{-1}$ with $A = 0.86$. On the other hand, we find richer dynamics for the MP, whose dynamics is characterized by $\Gamma_{MP}(\delta = 0) = 0.18\,\text{ns}^{-1}$ and $\eta_{MP}(\delta = 0) = 3.25\,\text{ns}^{-1}$ for the purely intercavity polariton condition, and $\Gamma_{MP}(\delta = -0.37) = 0.19\,\text{ns}^{-1}$ and $\eta_{MP}(\delta = -0.37) = 4.57\,\text{ns}^{-1}$ for the detuned case. In contrast to the LP, the weight of the exponentials for the MP is almost equally distributed as $A(\delta = 0) = 0.55$ and $A(\delta = -0.37) = 0.5$. The significantly slower dynamics of the MP shown in Fig. 3b is a consequence of the interplay between the two dynamical factors. Indeed, we observe that for very short times and linearizing the dynamical equations, $C_{LP}(t) \approx 1 - \Gamma_{LP}t$, whereas $C_{MP}(t) \approx 1 - [A\Gamma_{MP} + (1-A)\eta_{MP}]t = 1 - \gamma_{eff}t$. On resonance, $\delta = 0$, we obtain $\gamma_{eff} \approx 1.7\,\text{ns}^{-1}$ which is significantly smaller than the dominant decay rate of the LP. We attribute this effect to a competition between a reduced photon component of the MP, which suppresses the damping rate, and the presence of the dark-states reservoir of exciton laying close to the energy of the MP, which may favor non-radiative scattering and accelerated decay. We observe that the dynamics of the MP is slower than LP one, even though the MP energetically lies on top of the reservoir of dark-state excitons (see Fig. 3a).

The short-time dynamics reveal a significant change in the polariton behavior in the nanosecond range. Our observations of the

reflectance spectrum and the inherent three-level scheme allow us to understand this effect on the dynamics, arising as a consequence of the intercavity nature of the MP polariton. Finally, we see that the decay of the LP becomes slower as it approaches the energy of the ErB triplet state. This is evident from the asymptotic intensity value of the decay curve, which does not converge to the noise floor of the other curves, and from the stretching of its fast decay component. Here, the triplet state plays a role in two compatible ways: phosphorescence from its direct decay and intersystem crossing via scattering to the LP. These effects lead to the relatively small decreasing of the mono-exponential character of the LP for $\delta = -0.37$.

## Photoluminescence

To further investigate the nature of the polariton states we now turn our attention to the steady-state fluorescence. The system is pumped with a CW laser emitting at $\omega_p = 2.62\,\text{eV}$, which corresponds to an off-resonant excitation for all negative detunings. This means that we inject photons approximately equally in both cavities and produce excitons in the right cavity. For $\delta/eV = -0.37$, the MP is formed almost equally by both left and right cavity photons, while LP is formed predominantly by left cavity photons. Thus, we observe a higher fluorescence intensity from the MP than from the LP, as seen in Fig. 3c. In the Supplemental Information we show that as we decrease the detuning, the LP acquires a larger fraction of the photon and exciton component of the right cavity, which consistently conduces to an increased LP fluorescence.

However, as we approach the condition for $\delta/eV = 0$, the UP energy shifts until it matches the pump energy at normal incidence. We show in Fig. 3e, that the UP for $\delta/eV = 0$, is primarily formed by right cavity photons. Therefore, this configuration preferentially injects photons

into the right cavity. As we move towards the condition for pure inte-cavity polariton, the right cavity photon component of the MP decreases until vanishing, akin to what happens for conventional dark-state polaritons seen in atomic physics[20]. When the MP decouples from the right cavity photons, it becomes dark in the sense that it cannot be observed with a driving protocol that involves pumping photons into the right cavity. This is confirmed by the strong suppression of the fluorescence intensity shown in Fig. 3d, which demonstrates the transition of the MP to a pure intercavity polariton (see Eq. (2)).

The combination of all previous observations allows us to understand the significant change in the polariton behavior in the nanosecond range. On the one hand, the reflectance measurements and the inherent three-level scheme yield a direct signal of the polariton spectra, energy bands, linewidths, and Hopfield coefficients. On the other hand, the suppressed steady-state photoluminescence and slow dynamics of the MP reveal the reduction of its right cavity photon component. Therefore, these combined measurements demonstrate the emergence of an intercavity polariton with a tunable dispersion.

## Discussion

We have experimentally demonstrated the formation of intercavity polaritons composed by the admixture of photons and excitons sitting in physically separated optical cavities. The three energy level Λ-scheme underpinning the physics of this system implies, on resonance, the existence of an intercavity heavy polariton. We observed that the middle polariton branch transits smoothly to a pure intercavity polariton state with suppressed PL upon tweaking the photon-exciton detuning. This mechanism effectively decouples the intercavity polariton from free space leading to slower short-time dynamics. On resonance, the absence of one of the photon states in the composition of the pure intercavity polariton is confirmed by the steady-state fluorescence, whose intensity is suppressed under resonant pumping of the upper polariton branch. The interplay of the polaritons with the exciton reservoir needs to be taken into account as this introduces entropically favored scattering paths that may hamper the observation of more exotic physics. However, we stress that the short-time dynamics modification hints at that the emergence of a pure inter-cavity polariton may allow to overcome the effect of the reservoir. This motivates further studies on the interplay between the dark-state dynamics and the possible breakdown of quasiparticle picture[42,45,46].

Hybrid photonic-polariton systems are an ideal platform to explore many-body physics in multi-level coupled cavity systems. This shares analogy to interlayer excitons observed in stacked 2D materials, where the indirect character of the interlayer excitons gives rise to strongly interacting many-body states[47], long-lived exciton-polaritons[48,49], and new classes of polaritons[50,51]. Inspired by how twistronics in stacked 2D materials gave rise to non-conventional superconductivity[52], we envisage that the control of polariton dispersions can be useful to increase polariton correlations and facilitate the observation of non-trivial quantum phases in lattices-free polariton systems. Moreover, our strategy to generate slow-light does not compromise the photonic component of the polaritons, making it appealing for strongly correlated truly polariton states. The reduced in-plane propagation helps confining spatially polaritons without the additional complication of fabricating physical boundaries. This may lower the threshold needed for quantum phases transitions while maintaining the architecture of the system simple. Furthermore, the quantum entanglement between the photonic and molecular degrees of freedom may be unraveled by exploiting the spatially indirect character of the intercavity polariton.

## Methods
### Sample fabrication
The sample is composed by two vertically stacked Fabry-Pérot cavities fabricated on a glass substrate $10 \times 10$ mm² by multiple successive sputtering and spin-coating steps. The bottom 300-nm-thick Ag mirror was fabricated by magnetron sputtering operated at room temperature and a base pressure of approximately $10^{-6}$ Torr which, during deposition, is pressurized with argon flow to $3 \times 10^{-3}$ Torr. We deposited 99.99% purity Ag at a rate of 0.08 nm/s. The active layer of the first cavity is obtained starting from a solution of 25 mg of polyvinyl alcohol (PVA, Mowiol 44–88, 86.7–88.7% hydrolyzed, Mw ≈ 205,000 g/mol) dispersed in 1 mL of distilled water. Then, 9.8 mg of Erythrosin B (ErB, Sigma Aldrich with dye content > 90%) was added to the PVA/water solution, yielding a 0.5 M concentration. The ErB/PVA thin films were deposited by spin-coating at 2100 rpm using a 0.45 μm pore PTFE syringe filter, obtaining approximately 120 nm thickness. The first cavity is completed by fabricating a 20-nm-thick middle mirror on top of the active layer. The second cavity is formed by a Polymethyl methacrylate (PMMA, Mw ≈ 120,000 g/mol) layer embedded between the middle and top mirror. This layer is obtained starting from a 25 mg/mL solution of PMMA. The solution is spin-coated at 2600 rpm for 60 s using a 0.45 μm pore PTFE syringe filter and provides a slow thickness gradient centered around 140 nm. Using PMMA instead of PVA avoids the formation of micro-bubbles at the surface of the second cavity.

### Experimental set-up
Energy-momentum spectroscopy is performed in a homemade confocal Fourier optical microscope. Imaging the back-focal plane of a high numerical aperture microscope objective onto the entrance slit of a spectrograph (Kymera 328i, Andor) coupled to a sCMOS camera (Zyla 4.2P, Andor) is done by a Bertrand lens and provides direct access to the angular- and spectral-resolved reflectance. In our set-up, the sample is illuminated through a Plan Fluor 50×/0.8 NA objective (Nikon) with white light emitted by a halogen lamp. The focal spot full-width at half-maximum equals 14 μm. The collected light is dispersed by a diffraction grating (150 lines mm, blazed at 500 nm). Two linear polarizers in the excitation and collection path are used to select the s- or p- polarization. Angular-resolved reflectance is obtained by replacing the cavity with a commercial mirror, which allows to normalize the spectra reflected off the cavity at each angle with those obtained with the mirror at the corresponding angles. Angular-resolved steady-state fluorescence is measured by pumping the coupled cavity with a 473 nm continuous wave laser (Excelsior 473, Spectra Physics) coupled to the Fourier microscope in epi-illumination configuration and focused down to 1 μm. The laser power is attenuated to 30 mW to avoid local damaging of the sample. The pump laser is filtered by a 500 nm long pass filter and its polarization is selected by the same broadband linear polarizer used for reflectance. The measurements for different detunings are collected using the same integration time to ensure comparability.

Lifetime measurements are performed by a homemade time-correlated single-photon counting module coupled to the Fourier microscope. 100 ps laser pulses centered around 513 nm (LDH-P-C-520M, PicoQuant) are focused on the sample surface by the same epi-illumination path used for the steady-state fluorescence. We use a repetition rate of 10 MHz and an intensity such that the average photon count rate at the detector is always 0.04% of the excitation rate. The focus diameter is roughly 1 μm. The pump beam is filtered by a 525 nm long pass filter, while the appropriate 10 nm full-width at half-maximum bandpass filter selects the LP or MP normal incidence wavelength, for each detuning. The emitted light follows the same optical path as reflectance and steady-state fluorescence but is directed to a single-photon avalanche photodiode (MPD). The trigger from the laser driver (PDL 800-D, PicoQuant) and the signal from the detector is sent to a time-to-digital converter (Time Tagger 20, Swabian Instruments). All histograms are built with 100 ps binwidth. The instrument response function (IRF) has been measured in several ways to check the consistency of the result and ensure a reliable exponential fit for the short-time dynamics.

The relation between reflectance, fluorescence and decay measurements is guaranteed by the overlapping focus spots and the slow gradient slope of the PMMA layer thickness.

## Detuning-resolved measurements

In order to access experimentally a large set of detunings in a single sample, we designed the PMMA layer to exhibit a slow and almost linear gradient towards the peripheral zones. This radial gradient is controlled by the spin-coating rotation speed. A radial sample movement of one millimeter corresponds to increasing the PMMA thickness by approximately 50 nm. On the other hand, the ErB/PVA layer featured a constant thickness throughout the sample. The position of the focal spot is controlled by micrometer screws that permit shifting the sample in the focal plane of the microscope objective.

## Theoretical approach

To understand the emergence of our polariton states, a three-level system within the model of three quantum oscillators suffices, as we have demonstrate in the main text. It is convenient, however, to place our results into a general framework which may potentially include polariton-polariton interactions, finite density and temperature effects which although are out of the scope of our current study, may be relevant for future investigations. With this in mind, we complement our theoretical discussion with a Green's function formalism. We introduce the imaginary-time Green's function of the system, $\mathcal{G}_{\alpha,\beta}(\tau) = -\langle T_\tau[\hat{\psi}_\alpha(\tau)\hat{\psi}_\beta^\dagger(0)]\rangle$, where the subindices $\alpha, \beta$ correspond to the left/right cavity photon and exciton, respectively. The fields $\psi$ and $\psi^\dagger$ evolve according to Eq. (1). In terms of this formalism, the spectral function defined in the main text for the left cavity photon is $A^L(\omega) = -2\text{Im}\mathcal{G}_{11}(\omega)$. The Green's functions follows the Dyson equation $\mathcal{G}^{-1}(z) = [\mathcal{G}^{(0)}(z)]^{-1} - \Sigma(z)$, with the non-vanishing terms

$$\mathcal{G}_{11}^{(0)}(z) = \frac{1}{z - \omega_c^L(\theta)}, \quad \mathcal{G}_{22}^{(0)}(z) = \frac{1}{z - \omega_c^R(\theta)}$$
$$\mathcal{G}_{33}^{(0)}(z) = \frac{1}{z - \omega_X}, \quad (5)$$

and the self-energy given by

$$\Sigma_{12}(z) = \Sigma_{12}(z) = -t,$$
$$\Sigma_{23}(z) = \Sigma_{32}(z) = \Omega. \quad (6)$$

The Green's function can be obtained analytically in this case, in particular, we obtain

$$\mathcal{G}_{11}(z) = \frac{1}{\mathcal{G}_{11}^{(0)}(z) - t^2 \mathcal{G}_{11}^{(0)}(z - \Omega^2 \mathcal{G}_{33}^{(0)}(z))}. \quad (7)$$

After analytic continuation $z \to \omega + i0^+$, the energies of the polaritons are obtained by the position of the poles of the Green's function

$$\text{Re}[\mathcal{G}_{11}^{-1}(E)] = 0, \quad (8)$$

which has the three solutions coined lower, middle, and upper polariton. The residue

$$Z = \left(\frac{\partial \text{Re}[\mathcal{G}_{11}^{-1}(\omega)]}{\partial \omega}\right)^{-1}\Bigg|_{\omega = E}, \quad (9)$$

is connected to the Hopfield coefficients as $Z_i^\alpha = |C_\alpha^i|^2$. The dispersion of the cavity photons is given by $\omega_c^{(R/L)}(\mathbf{k}) = \frac{c}{n_c^{(R/L)}}\sqrt{k_z^2 + k_\parallel^2}$, where the incident light is along the $z$ axis, perpendicular to the cavity mirrors and the angle, $\theta$, is given by $k_\parallel = n_c^{(R/L)}\frac{\omega}{c}\sin\theta$. We should stress that this formalism leads to the same analytical expressions than those presented in the main text. The dissipation of the polariton branches is connected to the losses of the bare exciton and photon states via $\gamma_\alpha = \sum_i Z_\alpha^i \gamma_i$, where we use the Greek indices for the dressed polariton states and the latin indices for the bare exciton and photon states. For further details see Supplementary Information S1. $\Lambda$-scheme.

## Data availability

The experimental data generated in this study have been deposited in the public repository database [https://doi.org/10.5281/zenodo.10814748].

## Code availability

Data sets generated during the current study are available from the corresponding author on request.

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

## Acknowledgements

We thank Joel Yuen-Zhou for the critical reading to our manuscript and valuable discussions. G.P. acknowledges financial support from Grants UNAM DGAPA PAPIIT No. IN104522 and CONACyT projects 1564464 and 1098652. H.A.L.-G. acknowledges financial support from Grant UNAM DGAPA PAPIIT No. IA107023. A.C.G. acknowledges financial support from Grant UNAM DGAPA PAPIIT No. IA101923. C.L.O.-R. acknowledges financial support from Grant UNAM DGAPA PAPIIT IG100521, and IG101424. H.E.S. acknowledges support from DGTIC-UNAM under Pro-ject LANCAD-UNAM-DGTIC-419, DGAPA-UNAM Project PAPIIT No. IA106023, and CONAHCyT project CF-2023-I-468. A.C.-G., G.P. and H.E.S. acknowledge financial support of PIIF 2023. H.E.S., acknowledges Carlos Ernesto López Natarén for helping with the high-performance computing infrastructure. H.A.L.-G., A.C.-G., G.P. acknowledge support from Grant UNAM DGAPA PAPIME No. PE101223. R.A.R. acknowledges the scholarship provided by CONAHCyT (CVU. 1083224) for being part of Programa de Doctorado en Ciencias (Física) at UNAM.

## Author contributions

Y.A.G.-J., B.V., D.L.D., C.L.O.-R., H.A.L.-G., and G.P. performed the experiments. R.A., H.E.S., and A.C.-G. provided the theoretical analysis. H.A.L.-G., A.C.-G. and G.P. wrote the paper, with input from all authors. A.C.-G. and G. P. designed the project.

## Competing interests

The authors declare no competing interests
