## [Peer Review File · Nature Communications]

Intercavity polariton slows down dynamics in strongly coupled cavitiesReviewer #1 (Remarks to the Author):

García Jomaso et al. study two coupled cavities, one of which contains an active element (excitons) yielding a "polaritonic" cavity, while the other remains photonic exclusively.

Tuning of the resonant frequencies leads to the dispersions and Hopfield contents to vary, as is expected, with the Authors focusing on the resonant case between the left-cavity photons and the right-cavity excitons, in which case, the system decouples the right-cavity photons, furthermore over a large span of angles from the branch-emission, which keeps the polaritonic content of the resulting inter-cavity polaritons the same. This is a nice and noteworthy result. This leads the Authors to claim a flat-band for the dispersion of these polaritons.

Flat bands have been reported many times in polaritonic systems. Some literature is cited in the text but a more detailed coverage could be useful and interesting. There has also been several works on how polariton wavepacket dynamics (or closely related systems such as spin-orbit coupled BECs) is affected by effects of the dispersion, including negative-mass effects where the dispersion changes curvature, which would be usefully referred to, especially since in the particular implementation reported here, it is not entirely clear whether an actual flat-band, or merely a heavy-mass dispersion, is observed. From Eq. (3) and the figures, the latter seems to be a more accurate description. See in particular Fig. 3d as one of the alleged flat-bands. Would the Authors wish to contend that this is, in practice, close enough, they should compare the magnitude of their residual curvature with previous and other flat bands, would these be similarly flat parabolas. They themselves use the denomination of "quasi-flat dispersion" to describe other cases and they should apply this strict description to their main claim as well.

The text also makes confusing statements regarding the bright-dark character of their flat band. In one place they highlight that unlike conventional polaritons, their structure allows to retain a high photon fraction for heavy-mass (exciton-like) particles. In other places, they explain that as the flat band forms, its polaritons become dark. This is because the bright-photons are those from the left cavity while the dark-ones are from the right cavity. This is a fairly confusing way to describe the system (as both bright and dark simultaneously). It should also be noted that would the flat band becomes entirely dark, this would be a major problem for possible applications. It is mentioned that "while the dispersion can be further reduced, this would be at the expense of strongly suppressing the photonic component" so entering the flat-band regime appears detrimental (as one needs to make measurements on the sought properties of strong-correlations, slow light, etc.) Darkness is, however, a problem from the current excitation scheme (that comes from the right-cavity's upper polariton), not of the flat-band polaritons themselves, which could be excited from their left-photons, so I believe the Authors did not describe their system to its advantage.

It is also mentioned that "inter-cavity polaritons open up the quantum tomography protocols for local and independent measurements of the photonic and molecular degrees of freedom" which is definitely interesting, but should be clarified. From the calculated spectral functions, it seems to be assumed that the left cavity mode is the one being observed, which from how the structure is designed, sounds reasonable. How one could, however, observe the other cavity in isolation, let alone its molecular component?

Finally, the theoretical treatment is performed through imaginary time Green functions. It should be explained why this sophisticated technique is used whereas a simple three-oscillator model would provide a straightforward and essentially complete description. Unlike what is claimed, the fits are also not particularly good, even at the qualitative level: they do not capture the correct curvatures, see, e.g., Fig. 2c: the polaritonic flattening of the LP is not observed in the experimental data. Would a quadratic (no strong-coupling) fit match better in this case? Or are fitting parameters constrained in a way that explains such discrepancies?

Overall, this is a nice work with various noteworthy features in a possibly fruitful new type (and at room-temperature) cavity, which could open innovative routes towards strongly-correlated polaritonic phases. The claims should however be better supported, contrasted more explicitly to other flat-bands regarding their flatness and still requiring revisions for various clarifications.

Reviewer #2 (Remarks to the Author):

In the manuscript titled " Flatband slows down polariton dynamics in strongly coupled cavities", the authors investigate the realization of room-temperature slow light with Frenkel polaritons in a photonic architecture that comprises two vertically stacked cavities, employing excitons of Erithrosine B. The study systematically explores intercavity polaritons formed by an admixture of two cavity modes and an exciton mode as a function of detuning. Photoluminescence and fluorescence decay measurements are employed to discern the nature of polariton modes.

The article is carefully written. I appreciate the details provided in the method section. While various systems, such as electromagnetically induced transparency, have demonstrated the slow-light phenomenon, there is a notable interest in alternative and simpler approaches. However, I have reservations regarding the validity of the authors' primary claim – the efficacy of intercavity polaritons in generating "slow light." I suggest considering the publication if the authors convincingly establish the advantage of their architecture over the conventional exciton polariton scheme involving a single cavity mode and an exciton mode for slow-light generation. In this context, I offer the following comments for their consideration:

1. Exciton polaritons are slower than photon modes inherited from slow material responses. Within the relevant momentum window, exciton modes have almost zero dispersion compared to the cavity photon mode. Thus, the limitation to achieve slow polaritons is set by the cavity mode dispersion, which can be mitigated only by reducing the photon nature, as the authors have correctly pointed out in the case of two-level schemes. While the introduction of the Lambda scheme, involving another cavity mode coupled to the original two modes via tunneling, is noted, concerns persist about whether this scheme effectively addresses the challenges of achieving slow light using polariton systems. Specifically, in Figure 2a, the dispersion of MP appears small with exciton coefficients of approximately 0.8. Although tuning the lower-cavity mode to zero-detuning in Figure 2c reduces the exciton coefficient by only 0.15 to around 0.65, there is a noticeable increase in dispersion.
2. The claim of a 4-8 times reduction in dispersion with a 40% photonic component is highlighted. It would be beneficial to compare these results with the dispersion achieved in two-level schemes for polaritons with similar photonic components. A comparison, either theoretically or referencing relevant literature, would strengthen the assessment of the significance of their findings.
3. In the "short-time dynamics" section, the authors allude to a mechanism analogous to charge population trapping observed in atomic electromagnetically induced transparency. Expanding on this analogy would enhance the understanding of the proposed mechanism.
4. The authors attribute the absence of photoluminescence from MP at zero detuning to its decoupling from the right cavity. However, it is unclear how this affects the photoluminescence. What happens if the pumping energy is more such that UP energy does not match the pumping energy even in the case of zero detuning? Additionally, how does this differ from conventional two-mode systems, where the upper polariton does not show photoluminescence simply because its energy is higher than the exciton reservoir? This is crucial as the authors connect the dark nature of MP with the Lambda scheme in electromagnetically induced transparency.
5. Additional details on calculating Hopfield coefficients in the Supplementary Information (SI) would be helpful. Specifically, an explanation of how the involvement of two cavity modes, as opposed to a

single one, influences the analysis.

6. The author should mention that the eqn. (3) is obtained when the right cavity is detuned from the exciton, as noted in SI.

7. There is a typo in SI page 2: "... the MP does not loos its photonic character because it retains significant residue of the right cavity photon." It appears this should refer to the left cavity photon.

Reviewer #1

We thank Reviewer #1 for the careful revision of our manuscript as well as for the detailed feedback which has allowed us to significantly improve our manuscript. We are very happy to read that the Reviewer considers that this *“is a nice work with various noteworthy features in a possibly fruitful new type (and at room-temperature) cavity, which could open innovative routes towards strongly-correlated polaritonic phases”*. In the revised version we have attended to all the Reviewer’s comments, and we provide a detailed point-by-point reply to all his/her suggestions.

For clarity, we have highlighted in blue all the modifications to our manuscript.

1. *“Flat bands have been reported many times in polaritonic systems. Some literature is cited in the text, but a more detailed coverage could be useful and interesting. There has also been several works on how polariton wavepacket dynamics (or closely related systems such as spin-orbit coupled BECs) is affected by effects of the dispersion, including negative-mass effects where the dispersion changes curvature, which would be usefully referred to, especially since in the particular implementation reported here, it is not entirely clear whether an actual flat-band, or merely a heavy-mass dispersion, is observed. From Eq. (3) and the figures, the latter seems to be a more accurate description. See in particular Fig. 3d as one of the alleged flat-bands. Would the Authors wish to contend that this is, in practice, close enough, they should compare the magnitude of their residual curvature with previous and other flat bands, would these be similarly flat parabolas. They themselves use the denomination of “quasi-flat dispersion” to describe other cases and they should apply this strict description to their main claim as well.”*

The Reviewer has raised a very important point. As the Reviewer clearly identifies, flatbands and heavy-mass polaritons have recently attracted a significant amount of attention in various contexts. As also mentioned by the Reviewer, the ability to tune the dispersion of polaritons on demand is currently of great interest in many fields of physics, ranging from atomic physics to condensed matter and solid-state systems. We thank the Reviewer for highlighting the broad significance of our manuscript.

In the strict sense, we report the observation of slow-light, where the polariton dispersion is significantly reduced. Mathematically, this corresponds to the case of heavy-mass polaritons. Our setup allows us to reduce the polariton dispersion such that the width of the middle polariton band is comparable with its linewidth, which is also the case of most polariton systems where “true” flatbands are hard to be observed. On the other hand, any direct and quantitative comparison with other reported polariton flatbands is far from being straightforward for two main reasons: first, for typical reported experiments, the residual curvature is masked by the polariton linewidth, which makes hard to extract an effective mass. Second, the origin of the residual curvature is different for reported experiments. Typically, some residual curvature is found even for lattice symmetries that allow exact flatbands, and it is caused by experimental deviations from an idealized tight-binding Hamiltonian. As the Reviewer

suggests, we are prone to consider a band as flat if its residual curvature is smaller than its linewidth. We also stress that our flatband middle polariton extends over a considerable fraction of the light cone and it maintains a constant intensity across the angular range where it exists. For these reasons, we believe that the most accurate description is through direct comparison of the polariton bandwidth with its linewidth.

We agree with the Reviewer that this point should be completely clarified to avoid any misunderstanding.

Edits to the manuscript:

- In the revised version we have entirely re-structured the first two paragraphs of the Introduction to put the discussion of photonic and polariton flatbands in a broader context.
- In the discussion below Eq. 3 we have added a paragraph emphasizing the meaning of flatbands to accurately describe our system and the physical implications of slow-light.
- We discuss the physical origin of the reported residual curvature of lattice polaritons and our proposal.

2. "The text also makes confusing statements regarding the bright-dark character of their flat band. In one place they highlight that unlike conventional polaritons, their structure allows to retain a high photon fraction for heavy-mass (exciton-like) particles. In other places, they explain that as the flat band forms, its polaritons become dark. This is because the bright-photons are those from the left cavity while the dark-ones are from the right cavity. This is a fairly confusing way to describe the system (as both bright and dark simultaneously)."

We thank the Reviewer for pointing out this source of confusion. Indeed, the terminology can be puzzling as it merges concepts typically used in atomic physics with ideas found in condensed matter and solid-state physics. We have employed the term "dark-state polariton" inherited from the field of atomic physics where the propagation of slow-light is in the form of what is coined dark-state polariton. This is a bright mode in the sense that it does couple to photons, but it is referred as dark because it is infinitely long-lived (see also our calculations in Fig. 1 of the SI). We understand the source of the confusion as in condensed matter the term dark-state is used to describe an excitation that does not couple at all to the photon field. In our revised manuscript, we now use "dark-state polariton" to describe the former and "dark-state excitons" or "dark exciton reservoir" to the latter.

Edits to the manuscript:

- In the third paragraph of Short-time dynamics, we have added an entirely new discussion to avoid this confusion.

It should also be noted that would the flat band becomes entirely dark, this would be a major problem for possible applications. It is mentioned that " while the dispersion can be further reduced, this would be at the expense of strongly suppressing the photonic component" so entering the flat-band regime appears detrimental (as one needs to make measurements on the sought properties of strong-correlations, slow light, etc.) Darkness is, however, a problem from the current excitation scheme (that comes from the right-cavity's upper polariton), not of the flat-band polaritons themselves, which could be excited from their left-photons, so I believe the Authors did not describe their system to its advantage."

The fact that the slow-light becomes "dark" in the right cavity is not detrimental for its experimental observation nor is it for designing possible applications. As clarified in the previous point, our flatband is by definition a dark-state polariton which remains accessible with optical means in our configuration. Indeed, our experiments demonstrate that optical measurements that involve also the left cavity reveal properties of the flatband middle polariton: in Fig. 3a of the main text, lifetime measurements of the middle polariton are realized by pumping the system non-resonantly, i.e., without leveraging the upper polariton. It is also possible to conceive experiments which involve pumping the flatband polaritons directly from the left cavity.

Edits to the manuscript:

- On page 5, left column, second paragraph, we have remarked *dark-state polariton* to refer to polariton states avoiding the use of "dark-state" which may be confused with dark-state excitons. Throughout the text we now distinguish between dark-state polaritons and dark-excitons.
- In the paragraph before the Conclusions we have further elaborated on the ability to probe our slow-light by optical means.

3. "It is also mentioned that "inter-cavity polaritons open up the quantum tomography protocols for local and independent measurements of the photonic and molecular degrees of freedom" which is definitely interesting, but should be clarified. From the calculated spectral functions, it seems to be assumed that the left cavity mode is the one being observed, which from how the structure is designed, sounds reasonable. How one could, however, observe the other cavity in isolation, let alone its molecular component?"

This is indeed a very interesting point. The idea of quantum tomography is precisely as described by the Reviewer: to perform local and independent measurements of the photonic and matter component of the flatband polariton. While the photonic degree of freedom can be probed with optical measurements on the left cavity, the excitonic degree of freedom can be observed, for instance, with transport measurements on the right cavity.

Edits to the manuscript:

- We have extended the discussion on this point in the second paragraph of Page 2, left column.

“4. Finally, the theoretical treatment is performed through imaginary time Green functions. It should be explained why this sophisticated technique is used whereas a simple three-oscillator model would provide a straightforward and essentially complete description.

The Reviewer has addressed a very important point. Indeed, we employ a field-theory formalism to describe our system. While, in principle, a simple three-oscillator model is sufficient to explain the energy and Hopfield coefficients of the polariton branches, our formalism enables us to understand beyond these two aspects. For instance, it simplifies concepts such as the transparency window, provides an analytical expression for the mass of the slow-light polaritons and the range where the middle polariton remains as a heavy-mass polariton. Furthermore, it allows us to include the linewidths of the exciton and photons without appealing to a non-Hermitian Hamiltonian. From it, a direct relation to the spectral function of the modes is found which can be linked directly to experimental observables. We do agree with the Reviewer that this point should be further stressed in the manuscript.

Edits to the main text and SI:

- We have clarified this point in the second page of the main text (right column, third paragraph) adding a whole new paragraph.
- In the SI, we discuss this point in more detail.

Unlike what is claimed, the fits are also not particularly good, even at the qualitative level: they do not capture the correct curvatures, see, e.g., Fig. 2c: the polaritonic flattening of the LP is not observed in the experimental data. Would a quadratic (no strong-coupling) fit match better in this case? Or are fitting parameters constrained in a way that explains such discrepancies?”

We concede that at large angles there is some discrepancy between the theory and the experiment. We now stress about these differences in the revised manuscript.

However, we are strongly confident that our theoretical formalism provides us with the correct description of our observations. As the Reviewer correctly anticipated, in our theoretical description, we keep most of the parameters reasonably fixed for all our predictions. Particularly, the Rabi coupling and the tunneling coefficient are not supposed to vary as a function of the detuning. This leaves us with only the cavity detunings as adjustable free parameters. In principle, upon fitting independently each detuned reflectance measurement we could obtain a better agreement with the experiment. This includes the relatively small flattening of the LP band and its curvature.

We think that forcing the best possible fit by letting vary significantly a larger number of parameters or employing functions other than those derived from the polariton theory would conflict with the physical intuition. Therefore, we believe that the fit of the reflectance measurement set presented in the manuscript is the most accurate and honest.

In the revised version of the manuscript, we now detail on the point raised by the Reviewer.

Edits to the manuscript and SI:

- In the paragraph above Eq. 3 we explicitly mention these deviations.
- In the SI, we elaborate on how the parameters are constrained due to experimental conditions.

“5. Overall, this is a nice work with various noteworthy features in a possibly fruitful new type (and at room-temperature) cavity, which could open innovative routes towards strongly-correlated polaritonic phases. The claims should however be better supported, contrasted more explicitly to other flat-bands regarding their flatness and still requiring revisions for various clarifications.”

We are pleased to learn that the Reviewer believes our study can pave the way for the design of strongly correlated polaritonic phases. We also appreciate the constructive criticism provided by the Reviewer and have made our best efforts to address and incorporate all their suggestions into the revised version of the manuscript.

Reviewer #2

We thank Reviewer #2 for the detailed review of our work and appreciate the comments and suggestions provided, which encouraged us to improve our manuscript. We are pleased to note that the Reviewer finds our article carefully written and acknowledges the notable interest in alternative and simpler approaches to achieve slow-light. In the revised version of our manuscript, we have carefully addressed all points raised by the Reviewer.

For clarity, we have highlighted in blue all the modifications to our manuscript.

1. “Exciton polaritons are slower than photon modes inherited from slow material responses. Within the relevant momentum window, exciton modes have almost zero dispersion compared to the cavity photon mode. Thus, the limitation to achieve slow polaritons is set by the cavity mode dispersion, which can be mitigated only by reducing the photon nature, as the authors have correctly pointed out in the case of two-level schemes. While the introduction of the Lambda scheme, involving another cavity mode coupled to the original two modes via tunneling, is noted, concerns persist about whether this scheme effectively addresses the challenges of achieving slow light using polariton

systems. Specifically, in Figure 2a, the dispersion of MP appears small with exciton coefficients of approximately 0.8. Although tuning the lower-cavity mode to zero-detuning in Figure 2c reduces the exciton coefficient by only 0.15 to around 0.65, there is a noticeable increase in dispersion.”

We thank the Reviewer for raising this important point. We understand the concern of the Reviewer and we take the opportunity to better explain this point.

The Reviewer is entirely correct, the dispersion of the polaritons can only be reduced at the cost of their photonic component. This statement holds true for both two-level and three-level polariton systems. In fact, one should expect that for a specific set of parameters, the three-level scheme reproduces the two-level polaritons and recovers its physics. However, under different conditions, the three-level scheme allows us to explore uncharted regimes. We would like to stress the main differences between the two schemes:

a) On resonance, the middle polariton lies exactly at the energy of the bare excitons. In conventional two-level polaritons this is only possible for infinite positive or negative detuning, where polaritons are no longer a mixture of photons and excitons. Our three-level scheme is a very promising method for investigating the coupling between the middle polariton which is in resonance with the large reservoir of dark-state excitons.

b) It is true that in a two-level scheme the detuning of the polaritons can be taken to match any polariton mass achievable with the three-level scheme. However, once the detuning is fixed, the dispersion of two-level polariton is fixed, i.e., there are no longer free parameters to tune this dispersion. In contrast, because our three-level scheme includes an extra energy level, the resulting polariton dispersion can strongly differ from that of the single cavity. Therefore, the new detuning, δ_{LX} unveils a mechanism to efficiently manipulate dispersion.

c) The three-level scheme allows us to recover the physics of the two-level scheme while simultaneously retaining the pure spatial indirect character of the MP. This feature is unique as it unfolds opportunities to probe locally and independently the photonic and excitonic components of polaritons resembling the conventional ones.

d) Finally, the obvious difference of having three polariton bands instead of two allows designing more advanced excitation schemes, e.g., two-color resonant illumination, and provides the system with an extra channel for the decay dynamics.

We agree with the Reviewer that this point should be explained with more clarity. Therefore, we have added a new paragraph to our manuscript and dedicated an entirely new section to the Supplementary Information.

Edits to the manuscript and SI:

- We have added a whole new discussion in the main text (page 4) where we discuss this in detail.

- We added an entirely new section to the SI that includes an explicit comparison between the cases mentioned by the Reviewer, as well as new figures to completely clarify this point.

2. *“The claim of a 4-8 times reduction in dispersion with a 40% photonic component is highlighted. It would be beneficial to compare these results with the dispersion achieved in two-level schemes for polaritons with similar photonic components. A comparison, either theoretically or referencing relevant literature, would strengthen the assessment of the significance of their findings.”*

We agree with the Reviewer. To strengthen the benefits of the three-level scheme for purposes of slow-light a comparison with two-level polaritons is desired.

Edits to the manuscript and SI:

- We have added a new discussion to the main text (page 4, last paragraph) where we discuss this in detail.
- We have added a whole new Section of the SI to discuss this point and provide a direct and explicit comparison between the two cases.

3. *“In the “short-time dynamics” section, the authors allude to a mechanism analogous to charge population trapping observed in atomic electromagnetically induced transparency. Expanding on this analogy would enhance the understanding of the proposed mechanism.”*

We agree with the Reviewer. First, we apologize for the typo in our manuscript: the correct wording is coherent population trapping. This is a phenomenon that has been observed in literature by Alzetta et al (Nuovo Cimento Soc. Ital. Fis., B 36, 5 (1976)) and that has subsequently been linked to electromagnetically induced transparency. In Rev. Mod. Phys., Vol. 77, No. 2, April 2005, the authors recognize that the physics behind EIT, dark-state and coherent population trapping is the same. In atomic physics, the group velocity of a light pulse travelling through an opaque atomic gas is strongly reduced under the condition of EIT. This is because the light pulse is converted, within the gas, into a dark-state polariton, whose decay probability is canceled via destructive quantum interference. In our system, as a three-level system, we obtain a state that also consists of a coherent superposition as in Eq. 3. In the revised version of the manuscript, we have moved this analogy in the discussion around Eq. 3 where it is more natural to be understood. In addition, we have added a couple of new references where comprehensive reviews of dark-states and coherent population trapping can be found.

Edits to the manuscript:

- We have moved the discussion of the coherent population trapping below Eq. 3 of the main text to better illustrate its analogy to EIT.

- We have added new references regarding this point.

4. *“The authors attribute the absence of photoluminescence from MP at zero detuning to its decoupling from the right cavity. However, it is unclear how this affects the photoluminescence.*

We thank the Reviewer for giving us the opportunity to clarify this important point. The photoluminescence from the middle polariton is affected by the energy level alignment in the Lambda-scheme. For detuning δ_{LX} different than zero, the middle polariton is not flat and the associated right photon Hopfield coefficient is finite. This provides the middle polariton with a direct decay channel into photons of the right cavity which are then efficiently outcoupled to the far-field and measured as a photoluminescence signal. Conversely, when $\delta_{LX} = 0$, this direct decay channel is suppressed resulting in a reduced photoluminescence signal. It is important to notice that the photoluminescence intensity we measured is not exactly zero. There is a reduction of roughly an order of magnitude in the intensity when comparing the color scale of Figs. 3(c) and (d).

Edits to the manuscript:

- We added a paragraph on page 5, before the section Conclusions and Outlook where we discuss this point.

What happens if the pumping energy is more such that UP energy does not match the pumping energy even in the case of zero detuning?

This would be an interesting experiment but falls outside our current experimental possibilities. First, we lack a tunable laser or other appropriate wavelengths. Second, a direct comparison with Fig. 3 will be probably unfair because changing the excitation wavelength implies a different absorption of the metallic mirrors and active layer. At the moment, it is unclear to us which the proper normalization of these measurements would be that could allow their direct comparison.

However, a first approach to this problem can be found in Fig. 3a of the main text. There, we measured the time-dependent middle polariton emission for non-resonant driving. We observed emission from the flatband MP upon pumping the system at an energy lower than the UP, which corresponds to a more equal electric field intensity distribution across the system.

Additionally, how does this differ from conventional two-mode systems, where the upper polariton does not show photoluminescence simply because its energy is higher than the exciton reservoir? This is crucial as the authors connect the dark nature of MP with the Lambda scheme in electromagnetically induced transparency. “

Contrary to the upper polariton in the two-mode schemes, for the explored regimes, the MP lies always below or at the energy of the dark exciton reservoir. Therefore, its photoluminescence is in principle expected. The effect of reduced photoluminescence we observe is a consequence of the interplay between the pumping scheme and the flatband condition of the middle polariton.

Edits to the manuscript:

- We added a paragraph on page 5, before the section Conclusions and Outlook where we stress this point.

5. "Additional details on calculating Hopfield coefficients in the Supplementary Information (SI) would be helpful. Specifically, an explanation of how the involvement of two cavity modes, as opposed to a single one, influences the analysis."

This is a crucial point and we thank the Reviewer for bringing it to our attention. Indeed, to better elucidate the differences in the physics of three-level polaritons compared to two-level ones, it is important to provide a more detailed explanation.

Edits to the manuscript and SI:

- As suggested by the Reviewer, we have added an entire new section in the SI where we specifically address this comment.

6. "The author should mention that the eqn. (3) is obtained when the right cavity is detuned from the exciton, as noted in SI."

We thank the Reviewer for mentioning this. As noted by the Reviewer, all of our experiments are performed when the right cavity is detuned from the exciton. In the revised version of the SI we have improved the discussion about this point.

Edits to the manuscript and SI

- In the main text the discussion around Eq. 3 has been expanded.
- In the SI we have also extended the discussion about this and fixed a typo.

7. "There is a typo in SI page 2: "... the MP does not lose its photonic character because it retains significant residue of the right cavity photon." It appears this should refer to the left cavity photon."

We thank the Reviewer for alerting us about this typo. She/He is completely right. We have corrected this mistake.

Edits to the manuscript and SI:

- We have corrected this typo in the SI

Reviewer #1 (Remarks to the Author):

In their resubmitted version and reply, Garcia Jomaso et al. attend all the comments, but not all of them satisfactorily.

They do not refer to quasi-flat dispersions anymore although they still contend that "'true" flatbands are hard to be observed" in polariton systems. They further write: "In the strict sense, we report the observation of slow-light, where the polariton dispersion is significantly reduced". Despite all those precautions in their reply, in the text itself, the title remains focused on "Flatbands" (first word) and their introduction still sets the mood as "flatbands have led to the achievement of several breakthroughs". Are those breakthroughs (such as unconventional superconductivity) possible with significantly reduced dispersions and heavy polaritons only? Or is a qualitative exactly-flat character of the band necessary for them? They further write in their reply: "we are prone to consider a band as flat if its residual curvature is smaller than its linewidth". This looks convenient but little clarifies if there is a paradigm shift from heavy polaritons to flat bands. At any rate, all these considerations of whether flat bands are flat or not should appear clearly in the text. In the revised version, it is instead buried in the middle of the text as follows: "It is essential to stress that, in the strict sense, the MP is a slow-light polariton with a band that depends on momentum." [meaning that it is not, in a "strict sense", a flat-band] "However, precisely on resonance, the linewidth of the MP state remains within its bandwidth for a wide angular range, thus, it can be regarded as a flat band" Then the authors further justify this by saying that similar practices have been adopted earlier (they also repeat twice this full passage, by the way). In my opinion, this is muddling the waters. This point should be addressed frontally, at the very beginning, clarifying whether residual curvature hidden in the dissipation entitles to discuss flat band physics or if "in a stricter sense" this limits the phenomenology to slow-light, heavy polaritons instead. In which case the title and focus on flatbands should be revised. If I understand the Authors' point correctly, they further say that dissipation is allowing or at least reinforcing the flat bands, which is not a point highlighted in the text (where the Lambda structure is given all the merit instead). My inquiry to compare their case to other claims also met with the disappointing reply that this is not straightforward. But why not? They can compare their residual curvature to broadening ratios.

Then on their choice of theoretical modeling, of Green functions for a coupled three-oscillator system. They say that they need Green functions are needed ("enables us beyond") to describe linewidths "without appealing to a non-Hermitian Hamiltonian". I believe oscillators can easily be dissipative nowadays and feature a linewidth and that no Green function is needed to produce their theoretical results, especially the analytical ones. This even without making the Hamiltonian non-Hermitian, but in such a linear coupling anyway, would it even matter? They also say they can extract masses of the polaritons, but as a curvature of the dispersion, various types of masses are also easily extracted from straightforward oscillator models. I therefore do not think that their reply is correct and that there is physics beyond that of coupled three-oscillator models, i.e., this is all in the dispersion.

Finally, regarding the dark nature of their polaritons, they say that there is confusion because of different uses of the term "dark" by various communities: not coupling to light by condensed-matter physicists and as having a strong exciton component in atomic physics, as a result of which it is heavier as well as less bright. Is this understanding correct? The Authors don't make it so clear, they write: "it is referred as dark because it is infinitely long-lived". If it is infinitely long-lived, it is not clear how it can be bright. In the text, however, this becomes merely "a long-lived polariton state". I thus guess that my definition above is what they mean, in which case they should probably write "a long-lived polariton state with a SMALL photon component" (in upcases, the suggestion). Anyway, they do not lift completely such confusion, especially with sentences like: "As we move towards the condition for flat band polariton, the right cavity photon component of the MP decreases until vanishing exactly. Decoupling the MP from the right cavity photons [...] confirming the transition of the MP to a dark-state polariton." Their dark-polariton does decouple from light, like dark excitons, although it still retains itself a photon component from other optical modes within reach (two cavities). It is, after all,

a polariton! I believe I understand what the Authors achieve but I don't think they do a great job at providing a simple picture.

In my understanding, the Hopfield physics of three-oscillators can provide extended regions of very-heavy (not truly flat) dispersions, resulting in slow propagation phenomena, with furthermore great conveniences like remaining coupled to some type of light (left-cavity photons), thus remaining accessible to excitation and detection. This seems to me the great input of this work, along with its experimental realization. Dissipation could help "flatten" residual curvature but this does not seem central to their mechanism. This is dispersion-relation physics, not strongly-correlated, topological, flatband physics. So I remain unconvinced that their angle to the story is the correct one, and if the Authors wish to retain it because it sounds more important or appropriate (it would if they finally reach strongly-correlated phases of their introduction thanks to this flattening), I believe they should argue the case better, and in more transparency. I remain of the position that this is a nice work but that it is unsatisfactorily presented: with unclear, confusing, seemingly inflated results (flatbands from imaginary Green-function treatment). I would like their manuscript to be closer to the "strict sense" of what they are doing.

Reviewer #2 (Remarks to the Author):

I appreciate the authors for carefully answering the reviewers' questions and for the effort in improving the manuscript. The exploration of the three-level cavity system is intriguing, introducing an additional parameter to craft a unique polariton system with potential applications in realizing slow light. While I support the publication of the work, I would appreciate it if the authors could improve the clarity of the following two points for readers, either within the main manuscript or supplementary information SI.

1. Their measurement of short-time dynamics reveals that while the presence of a dark exciton reservoir close to the energy of the MP is expected to accelerate the decay, the damping rate is suppressed due to the "reduced" photonic component of MP for the $\delta=0$ case. Thus, this phenomenon appears unrelated to the vanishing curvature. In fact, the curvature for MP is notably higher than the $\delta=-0.37$ case (Fig. 2a-f). This discrepancy is especially noteworthy given the authors' assertion in the conclusion "that the short-dynamics modification hints at that slow-light possibly overcomes the effect of the reservoir."

2. The photoluminescence measurement primarily establishes that, at zero detuning, MP mode is decoupled from the right cavity. Thus, the photoluminescence measurement reveals how the mixture of left and right cavity photons changes with detuning. However, the extent to which this observation contributes to affirming the slowness of light, beyond validating the interplay of different cavity modes in their model, remains unclear.

REPLY TO REVIEWER 1

We are grateful to the Referee for the careful revision of our manuscript and for stressing some points that should be clarified further. We believe that their valuable comments have helped us to improve our manuscript. All of our edits to the manuscript are highlighted in blue in the revised version of the manuscript.

They do not refer to quasi-flat dispersions anymore although they still contend that "'true" flatbands are hard to be observed" in polariton systems. They further write: "In the strict sense, we report the observation of slow-light, where the polariton dispersion is significantly reduced". Despite all those precautions in their reply, in the text itself, the title remains focused on "Flatbands" (first word) and their introduction still sets the mood as "flatbands have led to the achievement of several breakthroughs". Are those breakthroughs (such as unconventional superconductivity) possible with significantly reduced dispersions and heavy polaritons only? Or is a qualitative exactly-flat character of the band necessary for them?

We agree with the Reviewer on the need to address this point earlier in the manuscript and shift the angle under which we present our results. In several condensed matter systems, the competition between suppressed kinetic energy and interactions is ultimately responsible for the emergence of various strongly correlated phases. Recent experimental observations hinge on the ability to tune and control the kinetic energy of electrons, which compete with electron-electron interactions. To obtain such phenomenology perfectly flat bands are not needed. In this sense, pursuing flatbands is more like a guideline to the research rather than its final goal.

Inspired by these condensed matter experiments, our proposal focuses on the ability to control and suppress polariton band curvature, which is directly related to the polariton kinetic energy. In the lambda-scheme we have realized, the polariton mass cannot be tuned to be infinite, i.e., perfectly flatbands are not possible.

To avoid any misunderstanding, we no longer use the word "flatband" in our manuscript. We have changed the title of the manuscript, which now reads as: **"Intercavity polariton slows down dynamics in strongly coupled cavities."** Moreover, the Abstract and Introduction have been rephrased to highlight that our motivation and actual results rely on the emergence of an intercavity polariton with a tuneable dispersion and spatially segregated photon and exciton components, rather than the production of perfect polariton flatbands.

In the main text, we have rephrased throughout the article the description of our results to avoid any wording that may suggest that our scheme produces perfect flatbands. Indeed, our main results in the manuscript are the ability to produce heavy polaritons with inter-cavity character and tuneable dispersion.

Changes to the manuscript

- The title of the manuscript no longer refers to "flatbands" and now reads as: **Intercavity polariton slows down dynamics in strongly coupled cavities.**
- The Abstract and Introduction have been rephrased to highlight that our motivation and actual results rely on the emergence of an intercavity polariton with tuneable dispersion and photon/exciton components, rather than the production of perfect polariton flatbands. We have rewritten completely the two first paragraphs of the Introduction.
- In the main text, we avoid any wording that may suggest that our scheme produces perfect flatbands. To avoid any misunderstanding, we no longer use the word "flatband" throughout the text.
- We have accordingly edited the Supplementary Information.

They further write in their reply: "we are prone to consider a band as flat if its residual curvature is smaller than its linewidth". This looks convenient but little clarifies if there is a paradigm shift from heavy polaritons to flat bands. At any rate, all these considerations of whether flat bands are flat or not should appear clearly in the text. In the revised version, it is instead buried in the middle of the text as follows: "It is essential to stress that, in the strict sense, the MP is a slow-light polariton with a band that depends on momentum." [meaning that it is not, in a "strict sense", a flat-band] "However, precisely on resonance, the linewidth of the MP state remains within its bandwidth for a wide angular range, thus, it can be regarded as a flat band" Then the authors further justify this by saying that similar practices have been adopted earlier (they also repeat twice this full passage, by the way). In my opinion, this is muddling the waters. This point should be addressed frontally, at the very beginning, clarifying whether residual curvature hidden in the dissipation entitles to discuss flat band physics or if "in a stricter sense" this limits the phenomenology to slow-light, heavy polaritons instead. In which case the title and focus on flatbands should be revised. If I understand

the Authors' point correctly, they further say that dissipation is allowing or at least re-enforcing the flat bands, which is not a point highlighted in the text (where the Lambda structure is given all the merit instead).

We reckon the point of the Reviewer and thank you for giving us the opportunity to clarify the relation between the lambda-scheme and dissipation. The lambda scheme does not need dissipation to provide a polariton band with reduced curvature. Even though it cannot provide an ideal flatband, it is sufficient to obtain and explain all the effects we discuss in our manuscript, namely the formation of a pure inter-cavity polariton, photon/exciton composition, reduced PL and slow dynamics. Dissipation is an extra ingredient which can hide residual curvature, thus giving the impression of an even flatter band. We mentioned it because it is always present in any experimental realization, but we agree that it is not strictly necessary for the discussion, nor is it a central point for our findings.

To avoid this source of confusion, as we have shifted the angle of the manuscript, we do not longer refer to the comparison between the linewidths and MP bandwidth.

Changes to the manuscript

- We have removed the discussion about residual curvatures and its interplay with dissipation to support flatband generation.

My inquiry to compare their case to other claims also met with the disappointing reply that this is not straightforward. But why not? They can compare their residual curvature to broadening ratios.

We agree that our previous reply was not entirely satisfactory (not even for us). However, we would like to comment that in experiments where polariton flatbands are reported (for instance Phys. Rev. Lett. 112, 116402, Phys. Rev. Lett. 116, 183902, Phys. Rev. Lett. 120, 097401, ACS Photonics 2020, 7, 2273–2281, Communications Physics 4, 39), while there is convincing evidence that flatbands can be produced, it is not straightforward to quantitatively extract the residual curvature from the reported figures. In any case, changing the angle under which we present our results by completely removing the emphasis on the flatness of our intercavity polariton band renders the need for a qualitative or quantitative comparison with literature results unnecessary.

Then on their choice of theoretical modeling, of Green functions for a coupled three-oscillator system. They say that they need Green functions are needed ("enables us beyond") to describe linewidths "without appealing to a non-Hermitian Hamiltonian". I believe oscillators can easily be dissipative nowadays and feature a linewidth and that no Green function is needed to produce their theoretical results, especially the analytical ones. This even without making the Hamiltonian non-Hermitian, but in such a linear coupling anyway, would it even matter? They also say they can extract masses of the polaritons, but as a curvature of the dispersion, various types of masses are also easily extracted from straightforward oscillator models. I therefore do not think that their reply is correct and that there is physics beyond that of coupled three-oscillator models, i.e., this is all in the dispersion.

We agree with the Reviewer on that one of the strengths of our manuscript is that our theoretical modeling of the intercavity polaritons introduced in our work can be effectively explained with three-level physics. In fact, the wave function in Eq. 2 and the schematic in Fig. 1 provide a straightforward and intuitive physical representation of our coupled cavities and are sufficient to understand our results.

Starting from the three quantum oscillators model, different equivalent theoretical approaches can be chosen. We employed Green's function formalism for three main reasons: a) The spectral function of the cavity photons, crucial for comparison with experimental data, can be readily obtained. b) This formalism has been widely used in the literature to study polariton physics, and we are very familiar with this approach. c) It represents a natural playground to describe, in future experiments, phenomenology beyond the one we showed in our manuscript.

We interpret Reviewer's point as Nature Communications being a journal where readers from various communities and who may not be familiar with Green's function theory meet. In view of their suggestion, we now avoid invoking explicitly the Green's function formalism in the main text. Without changing the essence of our theoretical results, nor the interpretation they provide to our experiments, we simplified the description of our system and results referring only to the simple three-level matrix shown in Eq. 1. We have relegated the Green's function formalism to the Supplementary Information and Methods. This trade-off gives a good balance between the community interested in the general understanding of our results and theoretical physicists interested in extending our studies including polariton-polariton interactions, driven-dissipative effects, among others.

We hope that these edits make our manuscript more accessible to the broad readership of Nature Communications.

Changes to the manuscript

- The Green's function formalism has been removed from the main text.
- We have modified the aesthetics of Eq. 1, such that it now becomes clear that the eigenmodes of the 3 x 3 matrix, corresponding to simple coupled quantum oscillators, provide the physical picture to describe the polariton modes in our system.
- We have rephrased the discussion around Eq. 2 to remark on how the energies and Hopfield coefficients can be obtained.
- When we introduce the spectral function, we now briefly describe its meaning and avoid the use of the Green's function.
- We relegated the theoretical description of the polaritons in terms of the Green's function to the Supplementary Information and Methods.

Finally, regarding the dark nature of their polaritons, they say that there is confusion because of different uses of the term "dark" by various communities: not coupling to light by condensed-matter physicists and as having a strong exciton component in atomic physics, as a result of which it is heavier as well as less bright. Is this understanding correct?

As the Reviewer remarks the source of the confusion is the different terminologies. In atomic gases, a dark-state polariton refers to the undamped propagation of light in a typically opaque medium. There, the propagation of the dark-state polariton is in the form of slow-light, which can possess an arbitrarily large or small photonic component. As the polariton propagates, it does not decay and it can only be measured as it exits the atomic cloud. It is important to emphasize that a dark-state polariton is, by definition, coupled to a light field. On the other hand, in condensed matter physics, a dark-state exciton is decoupled from the light field, as the Reviewer correctly points out.

In the manuscript we have modified Fig. 3a) to illustrate the inter-cavity polariton nature of our proposal and added a paragraph to connect with EIT experiments with atomic gases.

Changes to the manuscript

- We have modified Fig. 3a) and added a paragraph (before Sec. Short-time dynamics) to connect and refer the reader to the existing knowledge in atomic gases in the context of EIT and slow-light.
- We now clarify the connection between dark-state polariton in atomic gases and our proposal. (Paragraph before Conclusions and Outlook).

The Authors don't make it so clear, they write: "it is referred as dark because it is infinitely long-lived". If it is infinitely long-lived, it is not clear how it can be bright. In the text, however, this becomes merely "a long-lived polariton state". I thus guess that my definition above is what they mean, in which case they should probably write "a long-lived polariton state with a SMALL photon component" (in upcases, the suggestion). Anyway, they do not lift completely such confusion, especially with sentences like: "As we move towards the condition for flat band polariton, the right cavity photon component of the MP decreases until vanishing exactly. Decoupling the MP from the right cavity photons [...] confirming the transition of the MP to a dark-state polariton." Their dark-polariton does decouple from light, like dark excitons, although it still retains itself a photon component from other optical modes within reach (two cavities). It is, after all, a polariton! I believe I understand what the Authors achieve but I don't think they do a great job at providing a simple picture.

We thank the Reviewer for bringing to our attention that this point has not been entirely clarified. As the Reviewer correctly identifies and emphasizes, our three polariton states are, by definition, coupled to light and are not infinitely long-lived. In our manuscript, we use the term "dark-state polariton" to describe a purely intercavity polariton state that is completely decoupled **only** from the right cavity photons, but it is still coupled to the left cavity. This state can be observed in reflectance measurements (when performed from the left cavity as we discuss in the Supplemental Information), but, because it lacks the right cavity component, it does not provide a photoluminescence signal. Since the three-level scheme in our setup is identical to the case in atomic gases, and the state in Eq. 2 is analogous to the "dark-state polariton" found in that literature, we have retained this terminology.

To address Reviewer's concern about the simple picture, we have further explained the conditions to obtain a pure inter-cavity polariton and their consequences in terms of the formation of a "dark-state polariton", that mimics the dark polariton found in atomic gases literature.

Changes to the manuscript

- We clarify that when the MP becomes a pure inter-cavity polariton it becomes dark under the driving protocol where photons are injected into the right cavity. See paragraph before Sec. Conclusions and Outlook
- To avoid confusion, throughout the main text we now emphasize the pure inter-cavity nature of the MP rather than focusing on its darkness. We only use this terminology once to connect to the existing experiments on Electromagnetically Induced Transparency and slow-light with atomic gases.
- We have rephrased the sentences mentioned by the Reviewer.

In my understanding, the Hopfield physics of three-oscillators can provide extended regions of very-heavy (not truly flat) dispersions, resulting in slow propagation phenomena, with furthermore great conveniences like remaining coupled to some type of light (left-cavity photons), thus remaining accessible to excitation and detection. This seems to me the great input of this work, along with its experimental realization. Dissipation could help "flatten" residual curvature but this does not seem central to their mechanism. This is dispersion-relation physics, not strongly-correlated, topological, flatband physics. So I remain unconvinced that their angle to the story is the correct one, and if the Authors wish to retain it because it sounds more important or appropriate (it would if they finally reach strongly-correlated phases of their introduction thanks to this flattening), I believe they should argue the case better, and in more transparency. I remain of the position that this is a nice work but that it is unsatisfactorily presented: with unclear, confusing, seemingly inflated results (flatbands from imaginary Green-function treatment). I would like their manuscript to be closer to the "strict sense" of what they are doing.

We thank the Reviewer for emphasising the potential relevance of our work. We would like to comment that the original angle of our manuscript, namely producing flat dispersions with our Lambda scheme, was intended to stress that the control and flattening of polariton bands can be achieved with lattice-free setups. Furthermore, the Green's function method is a standard formalism in the community, we have used it extensively in our previous studies where we accompanied this theory by a simple explanation of the underlying physics. Thus, we would like to stress that the original angle of the manuscript and the formalism were intended to present our results from that perspective, and we are disappointed to read that this was misinterpreted by the Reviewer. However, we value their comments and have therefore made every effort to avoid any potential misunderstanding and to write our results as transparently as possible.

Answers to all comments mentioned here are found in our previous point-by-point reply. We hope that the Reviewer finds the revised version of our manuscript now completely clear and suitable for Nature Communications.

REPLY TO REVIEWER 2

I appreciate the authors for carefully answering the reviewers' questions and for the effort in improving the manuscript. The exploration of the three-level cavity system is intriguing, introducing an additional parameter to craft a unique polariton system with potential applications in realizing slow light. While I support the publication of the work, I would appreciate it if the authors could improve the clarity of the following two points for readers, either within the main manuscript or supplementary information SI.

We thank the Reviewer for the second round of revision. We are very happy to read that the Reviewer supports publication in Nature Communications after some clarifications. All our edits to the manuscript are highlighted in blue in the revised version of the manuscript.

1. Their measurement of short-time dynamics reveals that while the presence of a dark exciton reservoir close to the energy of the MP is expected to accelerate the decay, the damping rate is suppressed due to the "reduced" photonic component of MP for the $\delta=0$ case. Thus, this phenomenon appears unrelated to the vanishing curvature. In fact, the curvature for MP is notably higher than the $\delta=-0.37$ case (Fig. 2a-f). This discrepancy is especially noteworthy given the authors' assertion in the conclusion "that the short-dynamics modification hints at that slow-light possibly overcomes the effect of the reservoir."

We thank the Reviewer for raising this point and for giving us the opportunity to further clarify the relation between the middle polariton band curvature and decay. On the one hand, the reduction of the middle polariton curvature increases the number of states lying at the exciton reservoir energy. That is, the flattening of the MP is related to the expected accelerated decay. On the other hand, the strategy we employed to flatten the band, i.e., the λ scheme, is directly accompanied by a reduction of the right cavity photon component. We refer to this as the MP becomes a pure intercavity polariton. The competition between these two effects is summarized by the modification of the decay which, therefore, is indirect evidence of the polariton suppressed curvature.

One can note in Fig. 2 three important points: A) the Hopfield coefficients of the MP state for $\delta=0$ remain constant over a large angular range, in contrast, for $\delta=-0.37$ we observe a dependence on the angle of incidence. B) The energy of the MP for $\delta=-0.37$ does not match the exciton reservoir energy. C) At $\delta=-0.37$ the MP has a right photon component that provides a direct decay channel for the exciton. Therefore, only at zero detuning we expect the above-mentioned competition.

In the revised version of the manuscript, we emphasise the role of the energy matching together with the suppressed MP dispersion.

Changes to the manuscript

- The sentence mentioned by the Reviewer now reads as: *"However, we stress that the short-time dynamics modification hints at that the emergence of a pure intercavity polariton may allow to overcome the effect of the reservoir"*
- At the beginning of the Short-time dynamics section we have rephrased the first sentence to avoid any confusion as: *"Now we focus on the effect of the pure intercavity nature of the MP"*.
- We now elaborate this point in the SI.

2. The photoluminescence measurement primarily establishes that, at zero detuning, MP mode is decoupled from the right cavity. Thus, the photoluminescence measurement reveals how the mixture of left and right cavity photons changes with detuning. However, the extent to which this observation contributes to affirming the slowness of light, beyond validating the interplay of different cavity modes in their model, remains unclear.

The Reviewer has raised again a very important point. The PL measurements allow us to understand the mixture of left and right cavity components, which demonstrates that the polariton state is of the form of Eq. 2. This is akin to the dark-state polariton in slow-light experiments. We now remark this important point in the main text.

Changes to the manuscript

- In the paragraph before "Conclusions and Outlook" we now explain this in more detail.

Reviewer #1 (Remarks to the Author):

In this revised version, the Authors took the final step of getting completely free from the hype and invoking exotic concepts and buzzwords of great marketing value but that hinder the actual physics. I find this version particularly clear and convincing. The quality of the experimental results, of course, remains as high as they were previously. Their reply is also very good but this time the important points also made it to the main text. This is therefore the type of scientific work that, I believe, will raise the standard of the field and become milestones thanks to their rigor, sobriety and clarity. I believe the current text is indeed a great addition to both the physics of engineered light-matter systems and the collection of Nature Comm. papers. I would only suggest the Authors to consider widening the impact to other platforms, considering for instance PRL 118:155301 2017 & PRL 121:055302 2018 in spin-orbit coupled BEC, where this type of dispersion engineering led to beautiful negative mass phenomenology, which I have no doubt is a future avenue for the powerful platform put forward by Garcia Jomaso et al.

Reviewer #2 (Remarks to the Author):

Please find the comments in the attached PDF.

I appreciate the authors' response. However, upon further review, I have identified several areas that still require clarification before I can endorse publication.

1. Firstly, the authors do not address my first comment. The summary of the measurement of short-time dynamics is that for $\delta=0$, the decay rate of LP is 3.25 ns^{-1} , and the effective decay rate for MP is $\gamma_{\text{eff}} \approx 1.7 \text{ ns}^{-1}$. That is, the decay rate for MP is smaller, in contrast to the expectation that the flattening of the MP would lead to accelerated decay. They attributed the lower decay rate to the reduction of the right cavity photon component. Therefore, their measurement of decay rate **does not support the flattening of MP, only supports** the reduction of the right cavity photon component in MP and hence **its intracavity nature**. Thus, I am in agreement with the statement in the conclusion: "... the short-time dynamics modification hints at the emergence of a pure intercavity polariton ..."

However, I do not understand why they claim in the review response, "The competition between these two effects is ... *indirect evidence of the polariton suppressed curvature*." They also claim inside the main text: "Therefore, we conclude that *the suppressed curvature of the MP* and its concomitant intercavity nature produces a significant effect in the polariton dynamics in the nanosecond range."

They seem to employ a circular argument in this line: (i) The measurement of short-time dynamics establishes a reduced component of the right cavity photon. The reduction of the right cavity photon in the $\delta=0$ case is expected from the lambda scheme. Thus, this measurement establishes the validity of the lambda scheme. (ii) The Lambda scheme should have yielded flat MP. Hence, from (i) and (ii), their measurement is indirect evidence of flat MP. This line of argument is either flawed or, at best, an overstatement to support an inflated claim.

They also have not addressed my earlier observation: "In fact, the curvature for MP is notably higher (in $\delta=0$ case) than the $\delta=-0.37$ case (Fig. 2a-f)". Is this observation wrong? If so, please state it clearly. If this observation is right, then the main claim of the paper is invalid.

2. The answer to the second comment is also inadequate. The authors essentially affirm my understanding that "the photoluminescence measurement primarily establishes that, at zero detuning, MP mode is decoupled from the right cavity. Thus, the photoluminescence measurement reveals how the mixture of left and right cavity photons changes with detuning." And, all they add in response is: "This is akin to the dark-state polariton in slow-light experiments." This is merely a restatement of similar texts in the manuscript, "Our findings suggest that reducing polariton group velocity is *akin* to generating slow-light in atomic gases ...", or "... the character of the pure intercavity polariton *reminds* of the dark-state polariton observed in slow-light experiments performed with atomic gases." The whole point of my question is that just alluding to resemblance does not explain the physics; it requires more explanation.
3. I also note the omission of several edits in the revised version that addressed previous comments, such as the comparison between three-level and two-level schemes, the explanation of dark-state polaritons, and proposals for quantum tomography protocols. The authors should clarify the rationale behind these exclusions.

4. Finally, I went through the assessment of Reviewer 1 and shared a similar opinion that their claim of the flat band is inflated. It is noteworthy that the measurements of short-time dynamics and photoluminescence do not provide any direct evidence of a flat band, and they fail to quantify the claimed flatness convincingly. I also do not understand why, if their “work can be effectively explained with three-level physics,” they do not first use the three-oscillator model, point out the inadequacy of this simpler method (if any), and then use a more sophisticated method such as Green’s function formalism.

They now seem to direct the focus of the paper away from flat band toward realizing intercavity polaritons. I still believe that the work is interesting because it experimentally demonstrates a new scheme to tune a cavity polariton further by employing another cavity mode located in a different spatial location. However, their abstract and introduction still do not accurately capture the essence of the paper. In particular, they should expand on why realizing intercavity polaritons is a significant result. Additionally, they should further explain the role/relevance of the right cavity mode in achieving the intercavity polariton mode between the left cavity mode and the exciton mode in the right cavity. They should further include a detailed explanation of a simple three-oscillator system, elaboration on key concepts such as electromagnetically induced transparency, slow-light experiments in atomic gases, etc.

In summary, revising the abstract and introduction and addressing the aforementioned points would strengthen the manuscript.

Reviewer # 1

We thank Reviewer # 1 for carefully reading our manuscript.

1. In this revised version, the Authors took the final step of getting completely free from the hype and invoking exotic concepts and buzzwords of great marketing value but that hinder the actual physics. I find this version particularly clear and convincing. The quality of the experimental results, of course, remains as high as they were previously. Their reply is also very good but this time the important points also made it to the main text. This is therefore the type of scientific work that, I believe, will raise the standard of the field and become milestones thanks to their rigor, sobriety and clarity. I believe the current text is indeed a great addition to both the physics of engineered light-matter systems and the collection of Nature Comm. papers.

We are really delighted to read that the Reviewer considers our manuscript to have the potential for becoming a milestone in the field. We are very happy to read that the Reviewer recognizes and appreciates our efforts to improve the quality of the manuscript and that now recommends publication in Nat. Comm.

2. I would only suggest the Authors to consider widening the impact to other platforms, considering for instance PRL 118:155301 2017 & PRL 121:055302 2018 in spin-orbit coupled BEC, where this type of dispersion engineering led to beautiful negative mass phenomenology.

We appreciate the Reviewer for bringing this to our attention. We have added a new paragraph to the Introduction, elaborating on this point, and have included the references mentioned.

Reviewer # 2

We are grateful to the Reviewer for the constructive comments on our manuscript. In the revised version, we have made our best effort to carefully address all their suggestions. This includes a revised Abstract, Introduction, an improved Fig. 1, several new paragraphs and clarifications in the main text, together with a new section in the Supplementary Information, with two additional figures. All the edits to the manuscript and SI are highlighted in blue.

1. *“Firstly, the authors do not address my first comment. The summary of the measurement of short-time dynamics is that for $\delta=0$, the decay rate of LP is 3.25 ns^{-1} , and the effective decay rate for MP is $\gamma_{\text{eff}} \approx 1.7 \text{ ns}^{-1}$. That is, the decay rate for MP is smaller, in contrast to the expectation that the flattening of the MP would lead to accelerated decay. They attributed the lower decay rate to the reduction of the right cavity photon component. Therefore, their measurement of decay rate does not support the flattening of MP, only supports the reduction of the right cavity photon component in MP and hence its intracavity nature. Thus, I am in agreement with the statement in the conclusion: “... the short-time dynamics modification hints at the emergence of a pure intercavity polariton ...”*

We thank the Reviewer for the careful and precise explanation of the short-dynamics of the MP. Indeed, as the Reviewer recognizes, a small right cavity photon component is expected to suppress the direct decay channel for the MP and confirms the excitation of a pure intercavity polariton.

2. *However, I do not understand why they claim in the review response, “The competition between these two effects is ... indirect evidence of the polariton suppressed curvature.” They also claim inside the main text: “Therefore, we conclude that the suppressed curvature of the MP and its concomitant intercavity nature produces a significant effect in the polariton dynamics in the nanosecond range.”*

They seem to employ a circular argument in this line: (i) The measurement of short-time dynamics establishes a reduced component of the right cavity photon. The reduction of the right cavity photon in the $\delta=0$ case is expected from the λ scheme. Thus, this measurement establishes the validity of the λ scheme. (ii) The λ scheme should have yielded flat MP. Hence, from (i) and (ii), their measurement is indirect evidence of flat MP. This line of argument is either flawed or, at best, an overstatement to support an inflated claim.

The Reviewer has raised an excellent point which should have been clarified better in our previous reply. Our main claim, i.e., the formation of a pure intercavity polariton, is based on three complementary experimental protocols: reflectance, photoluminescence, and short-time dynamics. The reflectance measurements yield a direct signal of the polariton spectra,

including energy bands, linewidths, and Hopfield coefficients. The MP photonic component is further revealed by the suppressed steady-state photoluminescence. Therefore, these results allow us to demonstrate the emergence of an intercavity polariton with a tunable dispersion. On the other hand, the fluorescence lifetime unveils a slower dynamic of the MP. As the Reviewer correctly remarks, with the reflectance and PL measurements already at hand, one can understand the faster dynamic as the consequence of a reduction of the right cavity photon component in the MP.

We acknowledge the Reviewer's perspective that claiming the short-time dynamics of the MP is itself a demonstration of the flatband nature of the MP band could be seen as a circular argument. Rather than providing indirect evidence of flatband polariton, the short-time dynamic reveals consistency among our three measurement protocols and supports our main claims on the intercavity nature of the MP.

Therefore, we agree with the Reviewer that the sentence mentioned above should have been elaborated better. We further stress that in the previous and present version of the manuscript we have removed any emphasis to the flatness of the MP.

Edits to the manuscript:

- We have modified the discussion around this point (page 5, right column highlighted in blue).
- In page 2, right column, last paragraph we added a new paragraph to introduce the three experimental protocols we use and explain how they complement each other.
- We added a whole new paragraph in page 6, left column (third paragraph) to summarize our measurements and conclude.

3. *They also have not addressed my earlier observation: "In fact, the curvature for MP is notably higher (in $\delta=0$ case) than the $\delta=-0.37$ case (Fig. 2a-f)". Is this observation wrong? If so, please state it clearly. If this observation is right, then the main claim of the paper is invalid.*

We thank the Reviewer for insisting on this point. To reply, let us remark on some points.

- a. First and most important: the main claim of the manuscript is the emergence of a pure intercavity polariton with tunable dispersion, not its flatness. Moreover, the dynamic interaction with the exciton reservoir relies on the fact that at zero detuning, the middle polariton lies on top of the bare exciton energy. On the other hand, for $\delta = -0.37$ the MP lies far below in energy.
- b. With very large detunings, one can produce polariton bands with much suppressed dispersions. However, this compromises the nature of the pure intercavity polariton and it is at the expense of its photonic component.
- c. For $\delta = -0.37$ and $\delta = 0.0$ we find that at normal incidence and small angles, the polariton masses are comparable (of the order of 5 times the mass of bare photons). We believe that the observation made by the Reviewer, i.e., *“the curvature for MP is notably higher”*, arises from a visual effect due to the dispersive character for $\delta = 0$ **and large angles, where the curvature of the MP is notably higher, as the Reviewer correctly remarks.**
- d. We would like to stress that at zero detuning, the intercavity polariton character is retained for a wide angular range (see Fig. 2 i).

In the revised version of the manuscript, we have now addressed this point explicitly. Furthermore, we have added a whole new section and two new figures to the Supplemental Information to discuss the role of the right cavity photon in more detail.

Edits to the manuscript:

- We have added two paragraphs to the main text of the manuscript to discuss this point (page 5, left column: second and fourth paragraph).
- In a first paragraph in blue on page 5, we discuss explicitly the point raised by the Reviewer about the curvature of the MP for the two values of parameters mentioned by the Reviewer.
- The second paragraph discusses the role of the right cavity in the formation of the intercavity polaritons.

- A whole new Section has been added to the SI to elaborate in detail about the point raised by Reviewer.
- Two new figures have been included to the SI to explain the interplay between the different relevant cavity detunings, the formation of intercavity polaritons, and their effective mass.

4. *“The answer to the second comment is also inadequate. The authors essentially affirm my understanding that “the photoluminescence measurement primarily establishes that, at zero detuning, MP mode is decoupled from the right cavity. Thus, the photoluminescence measurement reveals how the mixture of left and right cavity photons changes with detuning.” And, all they add in response is: “This is akin to the dark-state polariton in slow-light experiments.” This is merely a restatement of similar texts in the manuscript, “Our findings suggest that reducing polariton group velocity is akin to generating slow-light in atomic gases ...”, or “.. the character of the pure intercavity polariton reminds of the dark-state polariton observed in slow-light experiments performed with atomic gases.” The whole point of my question is that just alluding to resemblance does not explain the physics; it requires more explanation.”*

We appreciate the Reviewer’s comment as it gives us the opportunity to expand on this analogy. In the revised version of the manuscript, we have now modified the Introduction and Abstract to remark the generalities of the three-level system and its relevance to other fields. In addition, we have improved Fig. 1 in the main text, to present the three-level scheme in a general form. This way, we hope other communities in physics will find it useful and appealing to design exotic phases of light.

Edits to the manuscript:

- In page 2 (right column highlighted in blue), a new paragraph has been added to explain in more detail the physics of electromagnetically induced transparency and slow-light.
- We added a new paragraph to the Introduction to better focus our results and scope of the manuscript on the inherent three-level physics of our study.

- We have improved the presentation of Fig. 1 to give our manuscript the correct perspective right from the beginning.

5. *"I also note the omission of several edits in the revised version that addressed previous comments, such as the comparison between three-level and two-level schemes, the explanation of dark-state polaritons, and proposals for quantum tomography protocols. The authors should clarify the rationale behind these exclusions."*

- a. Please note that we have not removed the comparison between the three-level and two-level polaritons. Indeed, there was, and it is still there, a whole section in the Supplementary Information devoted to this comparison. We have further elaborated on this.
- b. We thank the Reviewer for alerting us about the mistaken exclusion of the sentences on the quantum tomography protocols. We have amended this point.
- c. We have further elaborated on the explanation on dark-state polaritons.

Edits to the manuscript:

- In the paragraph before Section *Intercavity polaritons*, we discuss the proposal on the quantum tomography protocols.
- In page 5, left column (third paragraph), we further stress the comparison of two-level vs three-level polaritons.
- A new paragraph on page 2, right column (in blue) discusses the physics of slow-light and the meaning of dark-state polaritons in atomic gases.

6. *"Finally, I went through the assessment of Reviewer 1 and shared a similar opinion that their claim of the flat band is inflated. It is noteworthy that the measurements of short-time dynamics and photoluminescence do not provide any direct evidence of a flat band, and they fail to quantify the claimed flatness convincingly."*

In an effort to attend the concerns of the Reviewers, in the previous communication we removed the discussion around perfect flatbands and

we focused the attention fully on the intercavity nature of our MP. Indeed, the comment of Reviewer # 1 after addressing their comments is: ***“I find this version particularly clear and convincing. The quality of the experimental results, of course, remains as high as they were previously. Their reply is also very good but this time the important points also made it to the main text. This is therefore the type of scientific work that, I believe, will raise the standard of the field and become milestones thanks to their rigor, sobriety and clarity.”***

In the revised version of the manuscript, we have tried to further highlight the storyline of our manuscript. In particular, we have modified the abstract and added a new paragraph to the Introduction, and improved Fig. 1 to stress the motivation and scope of our results.

Edits to the manuscript

- A new paragraph has been added to the Introduction to discuss the wide scope of Lambda schemes in optics and condensed matter.
- We rephrased some sentences of the Abstract to remark the Lambda-scheme as a platform to control states of light.
- Fig. 1 has been improved.

7. I also do not understand why, if their “work can be effectively explained with three-level physics,” they do not first use the three-oscillator model, point out the inadequacy of this simpler method (if any), and then use a more sophisticated method such as Green’s function formalism.”

Indeed, since our previous version of the manuscript we have entirely removed Green’s function formalism from the main text and only use a three-level quantum oscillator model. Without loss of generality, we introduced the spectral function in a different but equivalent way, relegating Green’s function formalism to the SI and to the Methods section. In the revised version, we mention in which cases the Green’s function formalism provides additional inputs, for instance for the study of polariton-polariton interactions, few and many-body effects, and physics far away from equilibrium. All these physical effects are available to be studied with our set-up.

Edits to the manuscript

- We have completely removed Green's function formalism from the main text. In Methods we briefly mention the scope of this formalism.

8. *"They now seem to direct the focus of the paper away from flat band toward realizing intercavity polaritons. I still believe that the work is interesting because it experimentally demonstrates a new scheme to tune a cavity polariton further by employing another cavity mode located in a different spatial location."*

We thank the Reviewer for stressing the interest of our work.

9. *"However, their abstract and introduction still do not accurately capture the essence of the paper. In particular, they should expand on why realizing intercavity polaritons is a significant result. Additionally, they should further explain the role/relevance of the right cavity mode in achieving the intercavity polariton mode between the left cavity mode and the exciton mode in the right cavity. They should further include a detailed explanation of a simple three-oscillator system, elaboration on key concepts such as electromagnetically induced transparency, slow-light experiments in atomic gases, etc"*

We thank the Reviewer for the constructive comments. In the revised version we have made our best effort to address all the comments of the Reviewer both in the main text and SI.

Summary Edits (Highlighted in blue in main text and SI)

- We have rephrased and added new sentences to the Abstract to clarify the scope of the article
- In the Introduction, we have added a new paragraph stressing the wide range of physical systems where three-level schemes can be exploited in optics.
- We improved the presentation of Fig. 1 to give it more generality.
- The quantum tomography brief discussion has been added.

- An entire new paragraph (second page, right column) has been added to connect with the well-established EIT and slow-light physics in atomic gases.
- Below this paragraph, we comment on our experimental measurements and how they give robustness to our claims. We also summarize our conclusions in page 6, left column (third paragraph).
- In page 5, left column (third paragraph), we further stress the comparison of two-level vs three-level polaritons.
- We discuss the role of the right cavity detuning (page 5, left column fourth paragraph).
- In Sec. Short-Time dynamics we improved the discussion about the physical implications of the short-time dynamics measurements (page 5, right column)..
- We added a new paragraph in Methods to introduce the Green's function formalism. (Note, that the Green's function formalism is not invoked at all in the main text).
- In the SI we have added a whole new Section to discuss the role of the right cavity detuning.
- Two new figures have been added to the SI and this section to illustrate the role of the right cavity detuning.

We hope that the Reviewer finds our revised version suitable for publication in Nature Communications.

Reviewer #2 (Remarks to the Author):

I appreciate the authors' effort in answering the reviewers' comments. The edits are appropriate leading to a more accurate interpretation of the data. I particularly appreciate the new section in the SI.

The authors state multiple times that the main claim is not about the flat band anymore: "First and most important: the main claim of the manuscript is the emergence of a pure intercavity polariton with tunable dispersion, not its flatness." This statement seems to contradict the assertion made in the abstract, "we realize room-temperature slow-light with intercavity Frenkel polaritons excited across two strongly coupled cavities." In the context of this study, slow-light and flat bands are nearly synonymous, and the data presented do not provide direct evidence of slow-light. So, I would modify this statement so that the community is not misled. I strongly believe this work demonstrates tuning intercavity polaritons, not slow-light.

While this reduces the novelty of the work to some extent, I still find this work interesting. With the suggested adjustment to the abstract, I would recommend the publication of the manuscript.

Reviewer #2

We thank the Reviewer #2 for the reading of our manuscript and are very happy to read that the Reviewer recommends publication in Nature Communications. We have attended their last recommendation and modified the sentence in the abstract as the Reviewer has suggested.